environmental science

humid tropics, slash-and-burn, agroforestry, alley-cropping, soil phosphorus

**Author for correspondence:**
Michael Hands
e-mail: mhands400@btinternet.com

# The search for a sustainable alternative to slash-and-burn agriculture in the World's rain forests: the Guama Model and its implementation

## Michael Hands[1,2]

[1]Department of Geography, University of Cambridge, Downing Place, Cambridge CB2 3EN, UK
[2]Inga Foundation, Higher Penhale, Lostwithiel, Cornwall PL22 0HY, UK

MH, 0000-0002-0032-5521

This account describes the context, history and outcomes of a series of studies into the ecology of slash-and-burn (S-B) agriculture in the world's humid tropics. These studies, which began in the mid-1980s, identified promising lines of research and continued to field trials, in Central America, of candidate agricultural systems as possible sustainable alternatives to the practice. The only system to emerge from 7 years' comparative trial with any promise of sustainability, in this context, was the agroforestry technique known as alley-cropping; but only with trees of the genus *Inga*. Inga alley-cropping then underwent field trials with subsistence farming families in northern Honduras. The system was aimed at the twin objectives of achieving food security in basic grains, on minimal inputs, and of providing the means of eliminating further S-B in the region. Since then, Inga alley-cropping has become the heart of a sustainable and integrated rural livelihood model (the Guama Model) which is being implemented successfully in northern Honduras with some 300 families. These families had been attempting to subsist on a few hectares of land degraded by decades or centuries of S-B. The development of Inga alley-cropping, supplemented by rock phosphate and other mineral supplements, as a sustainable subsistence and cash crop alternative means that land previously being held in reserve for subsequent S-B operations can now be planted to permanent forms of agroforestry. Entire landscapes can be re-greened by productive agroforest vegetation. Achieving this at scale will require the investment of huge extension effort and funds. However, the environmental, social and economic returns are also huge; and they are sustainable. In this programme, we are seeing the vitality and goodwill of

hundreds of families focussed on the raising, planting and management of trees in ways that feed the living organisms of the soil and, hence, feed themselves. In so cheerfully planting out their own futures, they plant and reshape the future of their own country. Replicating this at scale, as Rattan Lal outlines below, could reshape the future of this planet. In the mid-1980s, progress on sustainable alternatives to S-B, especially in rain forests, was frustrated by a lack of conclusiveness in the literature as to why soil fertility fails so rapidly post-burn; but also by a degree of contradiction on the impacts of the burn on certain plant nutrients. Hands (Hands 1988 The ecology of shifting cultivation. MSc thesis, University of Cambridge) concentrated on the role of soil phosphorus and attempted to resolve these contradictions. The Cambridge Alley-cropping Projects (1988–2002) continued this theme and threw light on the question of sustainable food production in rain forest environments.

# 1. Introduction

When this work began, my driving concern was for both the survival of the world's remaining rain forests and the well-being of those millions of families who were trapped in poverty and food insecurity by a widely failing agricultural technique. Considering the scale of the social and environmental problem, it seemed almost inconceivable that no alternative existed, or could exist. Little was said at that time about climate change and the impact of slash-and-burn (S-B) practices on the atmosphere.

The Brundtland Report of 1987, p. 14 spoke of its main theme:

'The downward spiral of poverty and environmental degradation is a waste of opportunities and resources. In particular, it is a waste of human resources'.

The Earth Summit of Rio in 1992 laid emphasis on Sustainable Land Management and Rehabilitation. The eight Millennium Development Goals 8 years later included the commitments: '… to eradicate extreme poverty and hunger … to ensure environmental sustainability; etc…'. There was, by then, no shortage of good intentions.

In 2014, the United Nations (UN) published its report on the Right to Food and the importance of agroecology in this context [1] and, in 2017, the UN Committee on Global Food Security (CFS-44) pursued the theme of sustainable forestry (including agroforestry) in the context of food security and nutrition [2]. Cairns [3,4] adds the plea that indigenous peoples and practices in rain forests be taken more seriously. Pollini [5], in a more policy-oriented overview, concludes that the voices of S-B smallholders should be heard in this context.

Rattan Lal has researched and published in the field of healthy soil for decades. His most recent overview [6, p. 8] extrapolates from present degradation to restoration, via re-greening, by the end of this century. His conclusion, which includes relevant calculations, can hardly be bettered:

'Adoption of restorative land-use and site-specific best management practices, which conserve soil and water and strengthen elemental cycling, can create a positive soil/ecosystem carbon budget and sequester atmospheric $CO_2$ as soil organic and inorganic carbon. The cumulative technical potential of carbon sequestration at 178 Pg in soil and 155 Pg in vegetation between 2020 and 2100 can create a drawdown of atmospheric $CO_2$ by 157 ppm'.

Following the Rio+20 Summit in 2012, there emerged the 2015 Sustainable Development Summit at which 17 goals were adopted. Lal, here, links UN Sustainable Development Goals (SDGs) 1, 2, 6, 13 and 15 directly to 'Soil health and functionality'.

Here, I show that the sustainable rural livelihood for the humid tropics described here, the Guama Model, addresses positively, directly or indirectly, at least, 10 of the 17 UN SDGs while having no negative impact on the remaining seven. This is only possible because, as I hope to demonstrate, it is remedial of past environmental damage and regenerative of landscape and livelihoods.

# 2. The context of slash-and-burn agriculture in the humid tropics

The seminal work in this field is: 'The Soil under Shifting Cultivation' by Nye & Greenland [7]. They prefaced the work thus:

'Over 200 million people, thinly scattered over 14 million square miles of the tropics, obtain the bulk of their food by the system of shifting cultivation. They form a little under 10% of the world's population, and are spread over more than 30% of its exploitable soils. Up to a century or so ago, shifting cultivation had no very serious effect on the farmland of the tropics, since the soil and vegetation were given adequate time to regenerate after a period of cropping'.

Nye & Greenland [7, p. v]

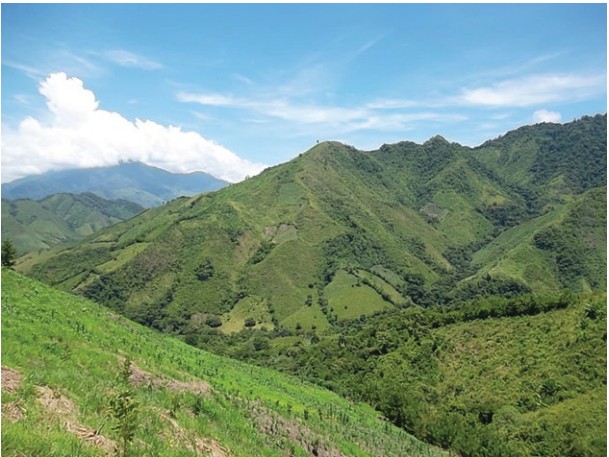

**Figure 1.** The Rio Viejo sub-catchment of the Cangrejal, Atlántida, Honduras. Impact on the landscape of a century of slash-and-burn.

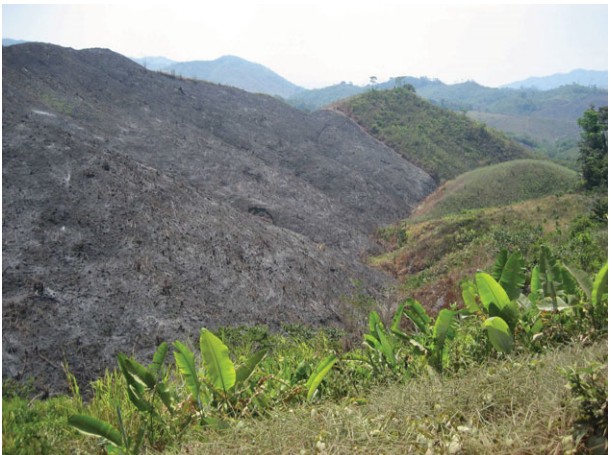

**Figure 2.** Near Capapan, Olancho, Honduras, 2008. Repeated slash-and-burn over decades has resulted in a very depleted soil, near-complete deforestation of the landscape and its replacement by invasive grasses.

The key phrase here is: '… the soil and vegetation given adequate time to regenerate…'.

We see, almost invariably today; marginalized families in Central America, compelled long ago to shorten fallow periods between S-B episodes on their (usually) inherited land; we see former forested land giving way, over whole landscapes, to fire-climax grassland (figures 1 and 2). This is no longer a 'system' as implied above, but a consumptive process in which the soil is ultimately exhausted of available plant nutrients.

UNESCO [8] estimated the global numbers as 200–300 million, which is probably an extrapolation of the 1960 figure. Taking into consideration population growth, and assuming one S-B operation per family, per year, it is arguable that this could now amount to 250–300 million families. Most subsistence farming families operate beneath their governments' 'radar' making even rough estimates difficult or impossible. What can be said, however, is that, over-and-above known human inputs (e.g. fuel), the surplus annual atmospheric $CO_2$ input attributed to 'land-use change' [9] is commensurate with this number of families S-B a hectare each of fire-climax grass/scrub. Whatever the accuracy of these estimates, the numbers are very large and the global environmental, economic and social problem is worthy of serious attention and remedy.

## 3. Chronology of the Cambridge studies and the Land for Life Program

1985–1987:  Desk studies in Cambridge; field studies in Costa Rica; laboratory work on rain forest and swidden soils in Cambridge.

1988:  Thesis submission.

1988–2002:  The four Cambridge Alley-cropping Projects in Sarapiquí and San Carlos, in the humid tropical lowlands of Costa Rica. These were followed by pilot implementation trials of Inga alley-cropping in northern Honduras [10].

2007:  A Charitable Trust is established in England to fund and implement the Guama Model in the humid tropics.

2008:  This non-governmental organization (NGO) is approved as a UK-registered charity.

2012:  The 10-year Land for Life Program is commenced in January 2012.

2014:  The Guama Model begins to be replicated in other countries in the World's humid and sub-humid tropics.

2020  The Guama Model is being replicated in 15 other humid and sub-humid tropical countries.

When these studies began, comprehensive reviews of the relevant literature already existed (Nye & Greenland [7]; Sanchez [11]; Jordan [12]); and it is not proposed to roam over the same ground here. However, since 1976, two more substantial studies of S-B impacts in rain forests have been published [13,14], together with the more-recent overview of Palm *et al.* [15]. The findings of those two field studies informed the Cambridge studies; and it is proposed here to highlight just the more important themes emerging by the mid-1980s. The context is S-B agriculture on the highly weathered, acid soils that are typical of rain forest environments.

Most authors agree on the short-term impacts of a burn on soil pH and cations [11,15–18], but the possible key role of phosphorus (P) has been markedly elusive and contradictory.

# 4. Models of sustainability in the humid tropics

Unlike most other subsistence agricultures, S-B cannot be intensified *per se*, as were others in the course of the Green Revolution from the 1960s onwards. Hands [19] concentrated on the ecology of S-B in rain forests and reviewed the relevant literature to date; but sought more widely for insights into, and models of, sustainability in this context. Leaving aside major transformation of the environment (e.g. Padi rice, raised fields, swamp cultures), scant few adaptations within indigenous rain forest agricultures served as indicators of sustainability. For this, there only remained the tropical rain forest (TRF) itself. The most biologically productive ecosystem on the planet provided the only useful model of sustainability; particularly in its nutrient ecology, root characteristics and symbioses.

The literature held strong indicators that the availability of P might be a key factor in the nutrient ecology of the more oligotrophic forests, typically on the highly weathered, leached and acid soils that dominate these environments [12,20,21]. Turner *et al.* [22], working on Barro Colorado Island, showed responses, within many tree species, to a resin-extractable P spectrum, but not within the community as a whole. Other tree species within the community must be obtaining their P from sources not tracked in this methodology and this could resurrect the old idea that mycorrhizal fungi might play a role in mediating the breakdown of soil organic forms of P via bacteria [23]. I argue below that dispersion/extraction techniques distort the P picture in the highly weathered soils (ultisols, oxisols) discussed here; and, further, that techniques using intact, or near-intact, soils and leaching tubes might provide greater clarity.

However, in the mid-1980s, the literature on S-B impacts was, at best, inconclusive regarding the possible role of P-availability in the sustainability, or non-sustainability, of crop production in the years following a burn. Moreover, two major studies (Uhl & Jordan [13]; Ewel *et al.* [14]) contradicted each other in this regard. I intend to resolve these contradictions in a more detailed treatment below, but note them here.

# 5. Losses of soil total phosphorus in the years following a burn

Hands [19] conducted a socio-economic enquiry among subsistence S-B farmers newly entering 'intervenido' (selectively logged) rain forest in Sarapiquí, Costa Rica during 1986, returning to Cambridge with soil samples from swidden sites and adjacent untouched forest. These also included the site of Ewel *et al.* [14] in which apparently anomalous findings regarding soil P had been reported.

Soil P fractionations, together with estimations of soil microbial-P, carried out in Cambridge, yielded useful insights; but it was the conceptually simpler determination of soil total P that finally

provided the needed breakthrough and the means of resolving confusing or contradictory findings regarding what has now emerged as the key role of P-availability in the ecology of S-B agriculture in this context.

This study revealed a *hitherto* unreported and substantial loss of soil total P in a burned plot of the Turrialba dystrandept soil of Ewel *et al.* [14]. The plot had been maintained free of all root ingress and plant establishment for some 7 years post-burn, sustaining a loss of 138 kg P ha$^{-1}$ in the top 20 cm. This is sufficient alone to explain the main driver behind shifting cultivation in this context [19], but this is not to claim that it is the only factor, as recent experience in Honduras bears witness (see below).

Ewel *et al.* [14] constructed budgets for all major plant nutrients, determining the nutrient contents of the major ecosystem components, including soil to 20 cm depth, of the rain forest which they felled, dried and burned. They give values for these components in the pre-slash forest, the pre-burn slash; the post-burn ash/soil and soil two weeks post-burn following rain. Hands [19], on checking the published values for the soil's starting P-content, discovered that their figure, when expressed as the usual ppm units, is given as impossibly low at 4 ppm. The value determined by Hands [19] was in the order of 1200 ppm (table 3). However, Hands' value for olsen-extractable soil P in that soil was 4 ppm which might explain the source of the mistake.

Had this not occurred, the researchers might have observed the complete loss of biomass P following the burn, as described in more detail for the San Juan site (table 2), and the way would have been clear for the development of a sustainable alternative. As it was, P-availability was not identified as a possible key factor in the non-sustainability of slash-and-burn until about 1986.

# 6. The Cambridge Alley-cropping Projects

This scenario laid the foundations for the (eventual) four Cambridge Alley-cropping Projects (The 'Cam' Projects, 1988–2002). These began, in collaboration with the University of Costa Rica, with a S-B operation in 2 ha of lowland secondary rain forest in San Carlos, Alajuela, together with an already-burned reserve site in Sarapiquí. The main project site, at San Juan, San Carlos comprised 28 randomized and replicated plots of 400 m$^2$ each, dedicated to examples of alley-cropping or bare-soil cropping. Half of the 16 main plots received applications of rock-P following the first baseline crop of maize and totalling 100 kg P ha$^{-1}$. Root ingress between plots was controlled by trenches. The cropping regime was deliberately taxing to the soil's ability to deliver nutrients; it comprised six-month continuous rotations of maize and beans; the classic basic grains of Central America. Alley-cropping tree species were, in the main plots, two locally recommended legume tree species, with (against local advice) eight species of *Inga* in adjacent plots. Other plots were established for destructive sampling, etc. Biomass production in all components, above and below-ground, was monitored directly or estimated by allometric means. The P-ecology of these systems was reported in Hands *et al.* [24]; and yields over 7 years, together with the performances of the various tree components were reported in Hands [25]. Interim project data are reported in [26–28]. Suffice it to say here that responses to the rock-P applications were immediate, significant and persistent in every ecosystem component, except *Gliricidia sepium*, over the entire 7 years' trial. The hypothesis regarding the relationship between rock-P and sustainability in basic grain production was upheld and indicated a clear distinction between the performance of this supplementary mineral and that reported for the readily soluble forms of fertilizer P in these humid tropical latosols. However, we did not carry out a direct comparison between the two forms on these plots.

Grain production in bare-soil (control) plots failed predictably within 2–3 years; that within alleys of *Erythrina fusca/Gliricidia sepium* failed within 3–4 years; and the only system to emerge from 7 years' trial with any promise of sustainability was alley-cropping with *Inga* spp. (Inga A-C); with the caveat that any sustainable system requires to be supplemented by rock-P.

# 7. The difference between Inga alley-cropping and alley-cropping as first conceived

Alley-cropping, as envisaged during the 1970s at IITA at Ibadan, took as its guiding premise the idea that the prunings should decompose in such a way that the nutrients liberated from their decomposition should be available for the crop sown immediately after the pruning in question. For this reason, the legume tree species had been chosen from those with smaller, more readily decomposable, leaflets:

*Leucaena leucocephala*, *Calliandra calothyrsus*, *Gliricidia sepium*, etc. However, fast-decomposing mulch offers little physical weed control while actually feeding the weeds as well as the crop [29,30].

Inga A-C dispenses with this completely and takes as its premise that the tough, recalcitrant mulch will simulate the sheltered conditions of the forest floor and will permit the fine-roots of both crop and trees to concentrate where they would be observed in the forest itself: in the very uppermost soil layers, the crop sown after pruning might derive some nutrient benefit from the decomposition of fine-roots and nodules but will derive very little from its immediate mulch cover [31]. It will, however, exploit nutrients liberated from previous prunings. In Inga A-C, in contrast to the original concept, we observe almost 100% weed control in the alleys.

# 8. Further confusion on the possible key role of soil phosphorus

One source of confusion regarding the role of P in these leached, acid tropical latosols stems from the failure to distinguish the possible leaching of P from an intact soil profile *in situ* and the same soil's P-sorption behaviour in the laboratory. This is outlined below, but noted here.

The findings outlined above of significant losses of soil total P over the years following a burn were confirmed on the San Juan site and are given in more detail in table 2.

# 9. The latter phases of the Cambridge projects

Subsequent Cam Projects saw the gradual shift of their centre-of-gravity from experimental and laboratory work in Costa Rica to field trials in northern Honduras where a pilot project with local institutions involved small Inga A-C plots with subsistence farming families in the buffer zone of the Pico Bonito National Park (PNPB; Cordillera Nombre de Dios); and in La Mosquitia. The present Land for Life Program derived from the experience and base facilities established at this time. The system was received favourably both by its target families and by those in the NGOs, etc., who would later be involved in its wider extension.

# 10. Implementing the Guama Model

The implementing NGO was founded in 2007 by former colleagues to raise awareness and funds as a UK-Registered charity (2008); thus to fund and implement the wider extension of Inga A-C. The system creates food security, and functions as a sustainable alternative to S-B in the world's rain forests.

Inga A-C lies at the heart of the Guama Model; an integrated rural livelihood model comprised of four agroforestry systems for food security in basic grains, cash crops, tree crops and valuable hardwoods.

This work is currently in year 9 of the 10-year Land for Life Program aimed originally at establishing the whole model, at a rate of 40 families per year, up to about 200; and to deliver proof-of-concept at a landscape scale. We currently (April 2020) have over 300 families involved and are being overwhelmed with demand from hundreds of others.

It was the 1986–1987 pilot project and subsequent Cambridge projects that provided the vital insights and proofs-of-concept that enabled the Guama Model to be developed. It is the ability of Inga A-C to enable a subsistence farming family to achieve food security and permanence on a formerly degraded site of their own choosing, and to end shifting agriculture, that distinguishes it from any other system known to us. The system will be described in greater detail below, but it is this ability of Inga A-C, aided by cheap mineral supplements, to restore the fertility of soils degraded by decades of S-B that places it at the heart of the Guama Model. We know of no other system capable of achieving this in this context. For example, the use of the forage legume *Mucuna pruriens*, is now widely rejected by farmers in northern Honduras where it is known as 'Frijol de abono' (fertilizer bean). There is now a widely held view that, on steep slopes, it is a cause of landslips; and, moreover, that it fails to control the invasive grass 'Caminador' (*Rottboellia cochinchinensis*) (M.R. Hands 1999, personal observation).

# 11. The Guama Model

The average subsistence family holding in northern Honduras is estimated to be about 8 ha (MOPAWI, O. Munguia 1995, personal communication), but many families have perhaps half of this area available.

This will commonly be comprised of steep, rock-strewn hillslopes, of which, perhaps, 0.75–1.25 ha would be S-B annually in a rotational sequence. The immense value of a sustainable alternative is that it frees land from this cycle and enables other permanent agroforestry land-uses to be established. The resulting Guama Model is comprised of the following four components:

(i) Inga A-C (5000 *Inga* trees ha$^{-1}$) for food security in basic grains;
(ii) Inga A-C, managed appropriately, for a range of cash crops (e.g. pepper, turmeric, etc.);
(iii) fruit trees: *Inga* in orchard configuration (400 ha$^{-1}$) as shade, as a source of naturally fixed nitrogen (N) and as companion trees for a range of fruiting tree crops (e.g. cacao, avocado, rambutan, citrus, etc.); and
(iv) reforestation: *Inga* as 'matrix' or 'framework' species for site-recapture; to be interplanted 6–12 months later with valuable tropical hardwoods.

Thus, the value of Inga A-C for food security is that it enables the other model components to be established.

## 12. What is Inga alley-cropping?

Young saplings of *Inga* (often *Inga edulis*, *Inga oerstediana*, etc.) are raised in a shaded nursery as near to the eventual site as water-availability permits (figures 3 and 4).

At the appropriate time (often two to three months), they are planted along the contours of the site, at 50 cm between trees, within the hedgerows; and at 4 m spacing between rows (figures 5 and 6). This 4 m wide space is the 'alley'; tree density is thus 5000 ha$^{-1}$. In the early phases of the present Land for Life Program, families were told that they could not expect any material benefit from the system for at least 2 years; i.e. until the *Inga* had closed its canopy and had achieved site-capture from the invasive grasses that almost invariably dominate these long-degraded sites (figure 7). At this point, and at the appropriate crop-planting time, the *Inga* are pruned to about 1.5 m in height, the foliage is stripped from the branches and mulched onto the soil surface (figure 8). The larger stems and branches are removed as a favourite domestic firewood, and the finer branches are laid along the up-slope side of the *Inga* stems as a barrier to any possible movement of organic material or soil. The mulch will settle at about 100–150 mm in depth within a few days. In Central America, the usual planting technique for basic grains is with a planting-stick (Espeque); however, it has become standard practice for all our 'Guamero' farmers, to plant under the shade of the *Inga* prior to pruning; then to prune and mulch. This achieves virtually 100% weed control; firstly, by shading; and secondly, by smothering under the mulch. Both basic grains possess enough reserves in the seed to push through the tough, thick mulch. Weed seeds (by definition, small) might germinate beneath the mulch, but, if they do, will lack the energy reserves to emerge through it. Once through, the crop will have a monopoly of light, soil moisture and nutrients (figure 9). The trees will begin to recover foliage by about five to six weeks, at which time the grain crop will be well-advanced. Following cropping (figure 10), the trees are left to recover their canopy until the following year. The process can be repeated annually, provided that minerals extracted in the grain crops are topped up from time-to-time and that gaps from occasional tree-mortality are refilled.

## 13. Fertility restoration on very degraded tropical latosols: one late lesson from the Cam Projects

At the San Juan site, at the 4-year post-burn stage, micro-plots were established within the 16 main experimental plots in order to test for responses to N (as ammonium nitrate) and, separately, to a mix of cations comprised of dolomitic lime (Ca, Mg and trace elements) and 'K-Mag' ($MgSO_4 + K_2SO_4$). This mixture was applied at varying levels within the replicated micro-plots. No responses to either treatment was observed in the basic grains during this or any subsequent year.

This contrasts markedly with recent experience in establishing Inga A-C in very degraded soils in the present Land for Life project in the humid tropical zone of northern Honduras. The term 'degraded' in this context refers to the outcomes of, at least, the following factors acting on the solum:

(i) loss of soil organic matter (SOM) from the original forest, together with its associated soil microbial biomass, owing to exposure and to the repeated liming effects of many S-B

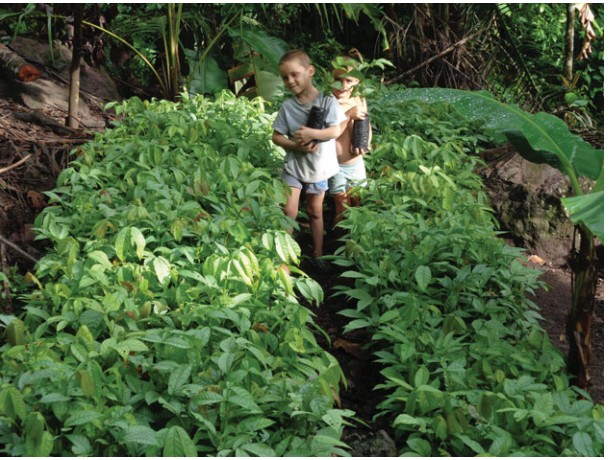

**Figure 3.** Cangrejal catchment. A family nursery shaded by palm-fronds. *Inga edulis* seedlings.

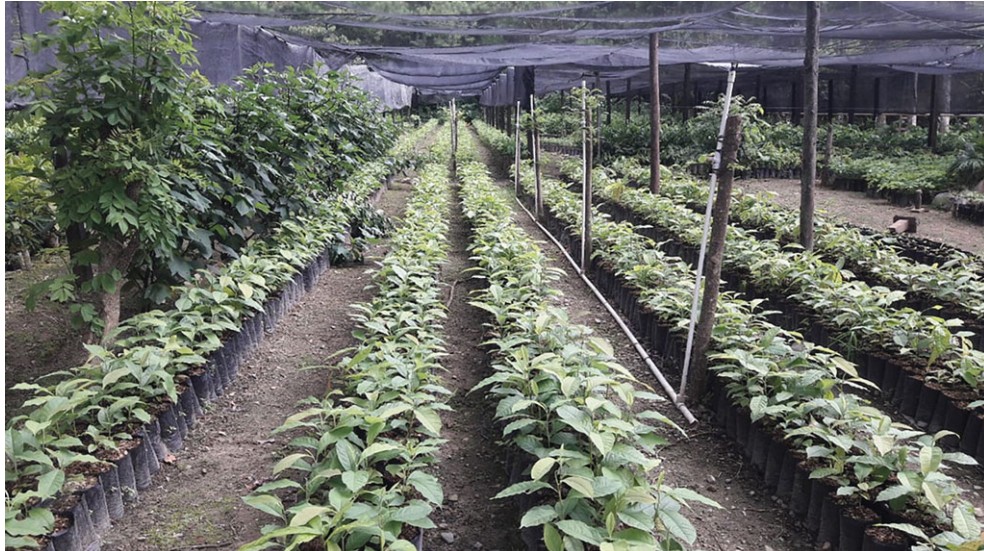

**Figure 4.** Part of the Land for Life nursery at Las Flores. Total capacity approximately 250 000 trees. Seedlings of the threatened, CITES-listed *Magnolia yoroconte*. This species is prized for fine timber and has largely disappeared from any location near human settlement.

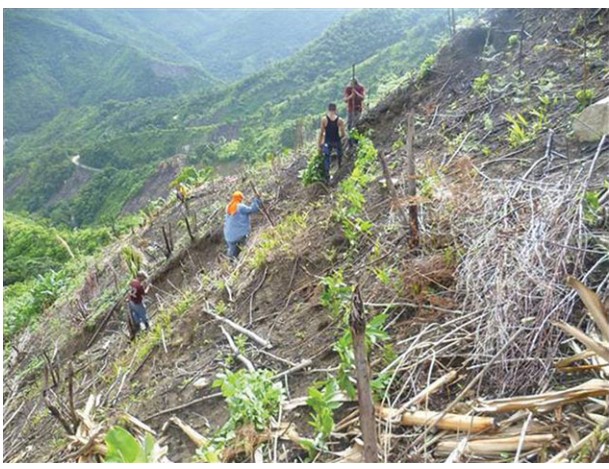

**Figure 5.** Cangrejal catchment. Planting saplings of *Inga edulis* along the contours of the site; 5000 will have been carried up from the nursery. Contours are found using an 'A-frame'.

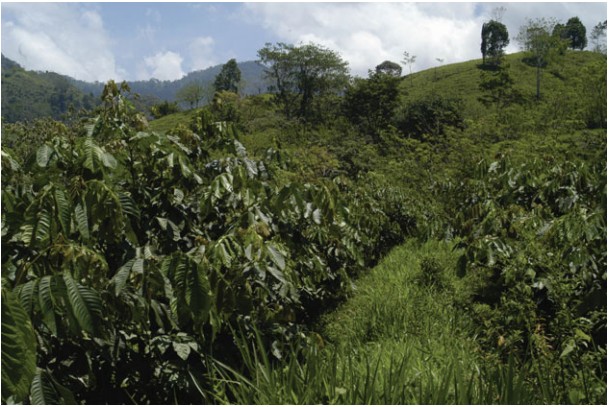

**Figure 6.** Las Flores. Cuero catchment, June 2013. *Inga edulis* hedgerows, seven months into site-capture from the grasses.

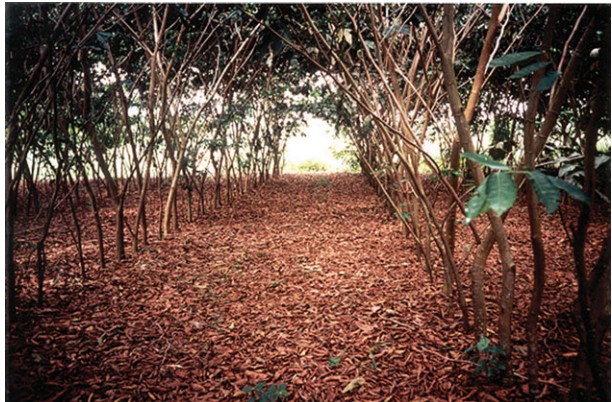

**Figure 7.** Cam Projects: San Juan site. Alley of *Inga edulis* at 2 years' growth and ready for the first pruning. No herbicides have been used. The aggressive grasses dominating the site have been eliminated by shade alone.

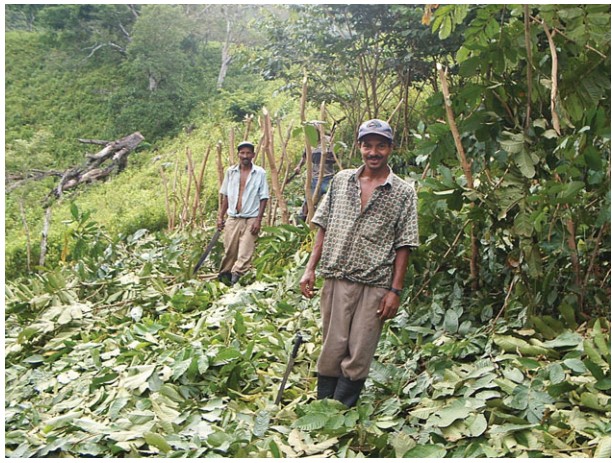

**Figure 8.** Cangrejal catchment. First pruning of an *Inga* alley plot. Deep, tough mulch will protect the soil surface layers from erosion or insolation. Weed-growth is suppressed and moisture retained beneath it.

operations. Loss of the P, N and S capital as contained in these components; together with the loss of cation exchange capacity and cations held in clay/SOM complexes;

(ii) loss of soluble plant nutrients from the ash of repeated S-B operations; especially, losses of P, N, S, K, Mg and Ca from the burned vegetation;

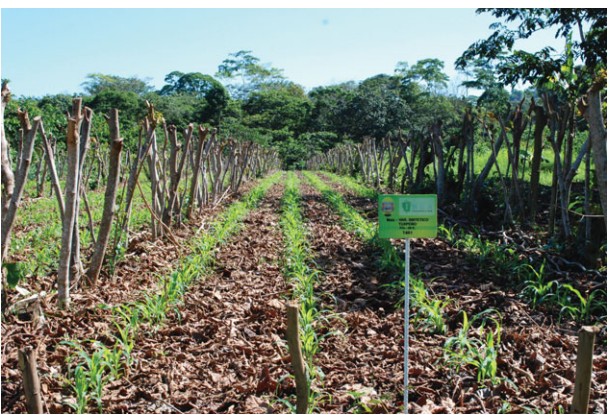

**Figure 9.** CURLA site, March 2015. Long-term experimental *Inga* alleys about two weeks after pruning and the sowing of maize. This is a demonstration facility and very much an ideal, flat location. The realities for subsistence farming families are very different. It does, however, show how the system looks and works. No herbicides are ever used in these plots. The trees (15 species) in the background were planted within a 'matrix' of *Inga* in 2000.

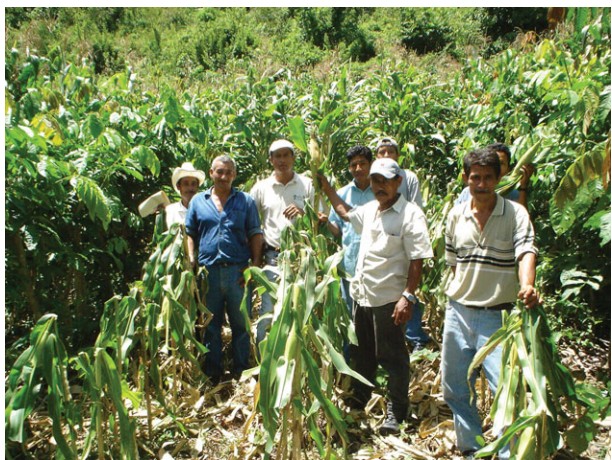

**Figure 10.** Cangrejal catchment. Alley-cropped maize doubled-over at maturity. The *Inga* is rapidly recovering its foliage. No weed control was needed.

(iii) simple loss of topsoil owing to erosion of soils unprotected by forest cover; and

(iv) loss of the moisture-holding capacity invested in natural levels of SOM beneath the original forest.

During year 3 of the Land for Life Program, we began to encounter examples of the *Inga* struggling to thrive, even though rock-P had been applied at planting out. In this case, the *Inga*/rock-P combination, although essential to soil restoration, in itself, had proved insufficient.

The response of the *Inga* to the addition of the dolomite/K-Mag mixture was dramatic. Chlorotic *Inga edulis* saplings at 1–1.5 m height grew powerfully to 3–4 m within three months. The chlorotic leaves themselves were not altered but were quickly shed. The new foliage, however, was a consistent and healthy dark green. Leaflet-area in these compound leaves was tripled.

Enquiry from this farmer, and others, revealed that these sites had been colonized and S-B from original rain forest at least 100 years ago. The only soil data, available from a site nearby, gave a soil pH of 3.4. It is not known how many S-B episodes had been carried out over this period, but they could easily have totalled 20, or more.

We attribute the difference in response between sites to their widely contrasting land-use history. The San Juan site in Costa Rica (soil pH 4.1) had supported secondary rain forest for at least 25 years prior to our burning operation. Moreover, the top 20 cm of soil at San Juan held no charcoal, although some fragments, together with primitive pottery shards, were found at about 25–30 cm depth (the limit of earthworm and termite activity on that site).

These findings proved to be of game-changing significance in the programme. The Rock-P, dolomite and K-Mag mixture (now known by the team as 'La Mezcla Magica') is now applied at all Inga A-C plantings. The family can now plant beans within the developing alleys instead of having to wait for a minimum of 2 years before first pruning and planting, as previously. This single factor has transformed the economics and nature of the technique and has removed, at a stroke, one of its few perceived demerits.

This combination of *Inga* and supplementary minerals (at approximately \$150 ha$^{-1}$) is the only system known in the region to be capable of restoring the fertility of such degraded soils. Long-term trials at the Centro Universitario Regional del Litoral Atlántico (CURLA), on the 20-year-old Cam *Inga* plots, indicate a requirement to 'top-up' these nutrients (approx. \$40–80 ha$^{-1}$) at roughly 4–5-year intervals. Smaller applications will produce a benefit but will need to be repeated at an earlier stage of managing the *Inga*.

Once the cash crop/tree crop components of the Guama Model become productive, these marginal costs become affordable; but the implication remains that adoption of the Model in highly degraded soils might need to be subsidized at its outset.

This is not 'liming', in the agricultural sense. The small application of dolomite, etc., is not an attempt to alter fundamental soil chemistry; nor does it need to be, even in these acid soils. The benefit lies in the addition of Ca, K and Mg cations as plant nutrients to the mulch and to the surface soil immediately beneath it. Mulching simulates the physical conditions of the forest floor; where fine-roots show a marked tendency, in acid soils, to concentrate in the sheltered upper few centimetres of soil and, in *Inga* alleys, into the lower layers of the mulch itself. On the Cam Project's Sarapiquí site, 31% of the fine-roots were concentrated in the top 2 cm of the soil of an *I. edulis* alley and 29% in the mulch itself; the remainder were found in the soil down to a depth of about 10 cm [25]. The cation supplements need only be held in this shallow zone. The thinking behind this use of the mixture is that the K-Mag delivers both cations in balance, thus avoiding chlorosis; whereas the slower release of Mg from the dolomite is a way of ensuring that there will always be a slight excess of this cation.

# 14. Weed control in the Inga alley-cropping system

Virtually 100% weed control is achieved in the Inga A-C system by two factors:

(i) firstly, shading by the dense *Inga* canopy; and
(ii) secondly, by smothering beneath tough, deep, recalcitrant mulch.

No agrochemicals have ever been required in over 25 years of managing Inga A-C and we know of no other system capable of controlling the invasive grass *Rottboellia cochinchinensis*. Shading is the key to this.

# 15. Pest control

In over 30 years of experience with Inga A-C, we have never observed defoliation by insects, or even a serious herbivory event. *Inga* are characterized by the possession of extra-floral nectaries located on the rachis, between the leaflets, of the compound leaf. *Inga* alleys abound with aggressive ants and, especially if near rain forest, predatory and parasitoid wasps. The *Inga* nectaries are fully formed and functional on even the smallest developing leaf; they are constantly the subject of attention from great numbers of ants and wasps (figure 11). The latter, if solitary, are thought both to paralyse and oviposit within the larvae; whereas, for both social insects, the larvae are a source of protein. The nectar is presumably a source of instant energy as they range and forage over both the *Inga* and crop plants in search of protein for the colony. Even in the case of alley-cropped *Passiflora edulis*; and where its famous herbivore *Heliconius* sp. butterflies are present, we observe very few larvae. Ants abound on the vines of the *Passiflora* which also possesses small nectaries at the base of the leaf.

During the growth of the first cropping cycle at the San Juan site, we observed a serious loss-of-condition in the bean plants. This was owing to an infestation of root-knot nematodes (*Meloidogyne* sp.). Similar root-knots were observed in the fine-roots of one of the trial alley-cropping legumes, *Erythrina fusca*. None were observed in any *Inga* sp., under trial at that, or any other, time; nor were they in the beans growing in the *Inga* alleys. The fine-roots of all *Inga* spp. in the trial became suberized soon after formation; this might have been a factor. No nematode problems have ever been observed in either crops or trees in Inga A-C during the 30 years since that event.

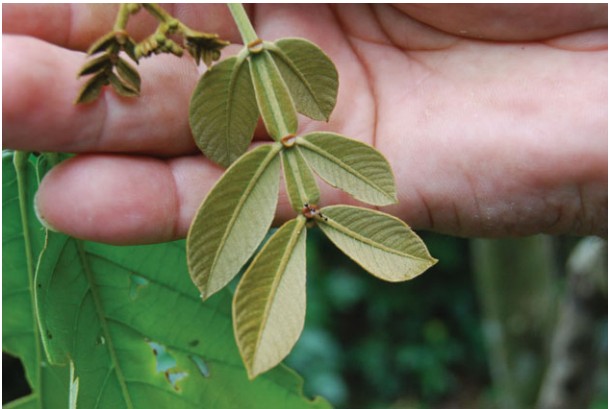

**Figure 11.** Developing leaf of Inga oerstediana. The nectaries are fully developed and functional. Two ants are active on the terminal nectary. Photo Mike Hands. Inga Foundation.

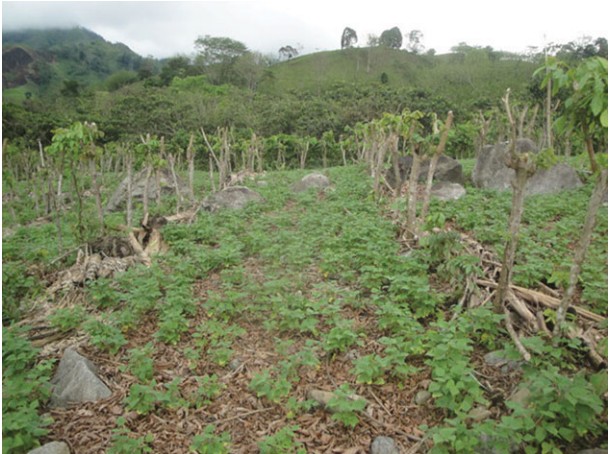

**Figure 12.** Las Flores. 26 March 2016. Beans in alleys of *Inga edulis*. Protected by the mulch, they and the soil survived 195 mm of rain in the storm of 21 March 2016. This was followed by nine weeks of drought and intense heat. The beans re-flowered and went on to produce a good crop. The soil beneath the mulch had retained enough moisture for the plants to do this.

## 16. Resilience in the face of climatic violence: two case histories

1. On 21 March 2016, the project areas in the Cordillera Nombre de Dios experienced a violent storm releasing 195 mm of rainfall over an 18 h period. All conventional S-B crop plantings were swept into the Caribbean and the Cangrejal river rose by over 10 m at Las Mangas. This, the most intense rainstorm to hit the area since Hurricane Mitch in October 1998, was followed by three months of drought and intense heat. Once we had regained access to our 'Guamero' farmers and our own demonstration farm at Las Flores (Valle de Cuero), we found that all the basic grain plantings (mainly beans) in the *Inga* alleys had survived (figure 12). Neither plants nor soil were lost. This we ascribe to at least three factors:

(i) the restoration of SOM, over some years, in the soil itself and, thus, of its ability to retain moisture;
(ii) the tough *Inga* mulch absorbing the force of the downpour and eliminating soil erosion; and
(iii) the pale, reflective nature of the upper surface of the mulch provided further physical protection against strong insolation, thus avoiding overheating and desiccation of the soil.

Both maize and bean plantings within the Inga A-C systems went on to yield crops.
2. We have seen something similar during 2019: one of our 'Guamero' farmers, Pablo, sowed beans in the shade of his Guama (*Inga*) alleys on 8 February of that year; a couple of days following the last rain to fall

in the valley until September. He then pruned the *Inga* and mulched the foliage on top of the soil and beans. He had chosen a quickly maturing variety called 'Cinquenteño' (50 days). Not a drop of rain fell for the whole of the growth period, after which he took in a heavy crop of beans. His neighbour, Victor, planted beans following an S-B operation at the same time. His plants did not even reach the flowering stage before they were lost. Victor now has 4000 *Inga* saplings and is totally convinced by his neighbour's *Inga* system. Since then, many similar stories have reached us from families in both valleys.

We now conclude that the system is resilient to the extent that, given some rainfall before planting, the soil-tree system retains much of it; and that a family with the *Inga* system would not have starved during the worst 'El Niño' event ever recorded to that date.

I can add, as a recent footnote, that these agroforestry systems are the only plantings known in northern Honduras to have emerged unscathed from the combined violence of two recent Central American hurricanes (Eta and Iota, November 2020). The relevance of, and need for, agroforest in subsistence farming in the present day is growing.

## 17. Tropical latosols 'fix' phosphate. Do tropical latosols leach phosphate?

In the mid-1980s, confusion and contradiction regarding the role of soil P in this context presented an obstacle to progress in understanding the marked failure in soil fertility which is commonly observed within perhaps 1–2 years of an S-B operation in the humid tropics. Without that understanding, the development of an alternative here could not take place. A few approaches were being suggested or trialled. Wilson & Kang [32] in Nigeria were already claiming alley-cropping with *Leucaena leucocephala* as a 'stable alternative to shifting cultivation'. Managed fallows and cultivation, in strip configuration, were being trialled at Pichis Palcazu in Peru. Uhl & Jordan [13] had concluded their study in the Rio Negro region of Venezuela with the statement that, although their data could not explain the rapid failure of crop production on their swidden, they speculated that it could be owing to a failure in P-availability.

De Las Salas & Folster [33], working in Colombia, reported apparent losses of soil P in the months following a burn. They discounted their own results thus: 'leaching of P need not be considered…'. It seems that laboratory findings of high sorption of P in these acid latosols (see also table 1) had caused scientific opinion to discount possible leaching through the soil profile.

This apparent contradiction was resolved by, among others, Sollins & Radulovitch [34], Radulovitch & Sollins [35] who demonstrated that soil micropore and micro-aggregate structure in a Costa Rican Inceptisol, very similar to the Ultisol of the San Juan site, permitted rain water to flow 'past' the micro-aggregates without chemically reacting with them. Since then, this factor of preferential flow has been demonstrated by others. This is to say that the behaviour of a soil *in situ* is radically different from that in a laboratory bottle in which the sesquioxide clays are fully dispersed and their sorption surfaces are exposed to the whole sorpable solute (potassium orthophosphate in the example given in table 1). The result is near-100% sorption; and the interpretation of results like these had caused widespread confusion. Table 1 is an example from the Cam Project's San Juan site, however, field data contradict this apparently decisive picture (table 2).

**Table 1.** The difference in the behaviour of a tropical latosol *in vitro* and *in situ*. (Sorption of added phosphate (P) in a tropical ultisol. San Juan site, San Carlos, Costa Rica (soil: pH 4.1). 1.0 g soil shaken; 1 h d$^{-1}$; 7 days: % sorption.)

| µg P added in 20 ml H$_2$O → | 375 | 750 | 1500 |
|---|---|---|---|
| % sorption | | | |
| soil depth (cm) ↓ | | | |
| 0–2 | 97.0 | 97.6 | 92.2 |
| 5–10 | 99.7 | 98.8 | 94.2 |
| 25–30 | 100.0 | 100.0 | 99.9 |

**Table 2.** The difference in the behaviour of a tropical latosol *in vitro* and *in situ*. (Soil total P concentration by layer; by year: Pre- and post-burn. San Juan site. Control (bare-soil) plots. Ultisol (pH 4.1). Soil total P: ppm dry soil (s.e. ±).)

| soil layer (cm depth) → | | 0–5 | 5–10 | 10–20 | 20–40 | 40–60 |
|---|---|---|---|---|---|---|
| treatment ↓ | | | | | | |
| pre-burn forest | (ppm soil) | 947 (15) | 836 (19) | 771 (11) | 683 (9) | 609 (16) |
| seven weeks post-burn | (ppm soil) | 933 (24) | 828 (4) | 769 (6) | 680 (10) | 616 (11) |
| 5 years post-burn | (ppm soil) | 848 (16) | 751 (12) | 667 (12) | 555 (17) | 484 (12) |

A number of field studies (e.g. Harcombe [36]) claim to have demonstrated that, for example, secondary succession in a TRF clearing, is not P-limited. Closer to the subject of this paper, Szott & Davey [37] state that *I. edulis* did not respond to phosphate applications on their Yurimaguas (Peru) ultisol. In both cases, the P was applied as triple superphosphate. Alley-cropped *I. edulis* at the San Juan site responded immediately and significantly to the application of P, as rock phosphate. The response persisted for the whole 7 years of the trial [25]. I suggest that, in the first two cases, the fertilizer P leached too rapidly to evoke a response in the vegetation.

In table 2, the soil's total P concentration, in all layers sampled, is nearly identical pre- and post-burn. Other data from that study suggest that the secondary forest canopy would have contained about 200 kg P ha$^{-1}$. If P-sorption were a factor in this soil, this P would have appeared in the upper soil layers following the burn. The data imply that all of it had leached during the seven weeks that had elapsed between samplings. Moreover, the 5-year data show significant losses in all layers sampled. Losses to 10 cm depth are only partially explicable by P-export in the grain crops; whereas those at greater depth can only have occurred via leaching.

# 18. Methodologies for determining total phosphorus-content of soil or plant material

Laboratory methodology can also distort the picture. Determinations of soil total P, soil organic P, etc. at that time, commonly involved the mineralization of the soil in oxidizing acids. Kjeldahl apparatus was widely used for this, giving rapid oxidation at high temperatures. Hands [19] attempted to determine soil microbial-P by the method of Brookes *et al.* [38], together with a fractionation of soil P components, following the method of Bowman & Cole [39].

In the latter procedure, sequential extraction with three reagents yielded three inorganic, three organic fractions and a residue. 'Organic' in this case being inferred from the difference in inorganic and total P in each extraction. As a check, whole soil was also digested. Despite great care in seiving, homogenization, etc., of the soil, the results were widely disparate. Exhaustive research suggested that the temperature of the oxidation might be a factor. The Kjeldahl equipment was abandoned and replaced by a heating block in which temperatures were progressively lowered. The oxidizing medium was a tri-acid mixture of perchloric, sulfuric and nitric acids. It was not until the digestion temperature was dropped below 160°C that the system seemed to stabilize. Not only did the standard P solutions emerge from the digestion unaltered, but the soil total P replicate values converged. The addition of $MgCl_2$ to the digest [40] did not resolve the problem. It seems likely that, at higher temperatures, some orthophosphate is converted to forms not detected by the Molybdenum Blue method of Murphy & Riley [41]. All further work was carried out at 130°C. Table 3 gives an example [19].

**Table 3.** The determination of soil total P-content. (Different methodologies; different results: Dystrandept, Turrialba, Costa Rica (site of Ewel *et al.* [14]). Soil layer: 10–20 cm. Soil total P by three methods: ppm (s.e. ±).)

| digestion method → | Kjeldahl | whole soil | sum of soil fractions |
|---|---|---|---|
| | tri-acid at approx. 400°C | tri-acid at 130°C | tri-acid at 130°C |
| site ↓ | | | |
| forest | 998 (±177) | 1227 (±21) | 1243 (±26) |
| clearing (7 years) | 993 (±129) | 1100 (±14) | 1132 (±20) |

In the Turrialba example, the high-temperature digestion has obliterated an important difference between the two adjacent sites. Losses of soil total P, to 20 cm depth, over this period amounted to 138 kg P ha$^{-1}$ [19]. Many contemporary studies were based on data derived from similar high-temperature digestions of soil or plant material.

All values for total P-content of all ecosystem components (e.g. grain, foliage, stems, soil, etc.) on the San Juan and Sarapiquí sites were determined by the low-temperature tri-acid method [24]. P determinations were carried out using an atomic absorbtion spectrophotometer and the Molybdenum Blue methodology of Murphy & Riley [41].

On the San Juan site, 5-year losses of soil total P to 20 cm depth were 164 ($\pm$21) kg ha$^{-1}$ and to 60 cm depth were 703 ($\pm$63) kg ha$^{-1}$ (M. R. Hands 1995, unpublished data).

Assuming the loss of 200 kg P ha$^{-1}$ from the original forest above-ground biomass, and assuming that a crop of maize in this context will extract 5 kg P ha$^{-1}$, this total ecosystem P-loss of approximately 360 kg P ha$^{-1}$, to 20 cm depth, is the equivalent of that phosphorus extracted (theoretically) by 72 maize crops; it will have derived from the more labile P fractions.

We tried to determine the fraction of soil P sustaining the majority of this loss; with inconclusive results. The strong suggestion from this work and that of Hands [18] is that most of it derive from the soil microbial biomass. On these bare-soil plots, inputs of organic matter from litterfall and root-turnover of the original forest have ceased; relatively small inputs are contributed by crop residues and weeds. However, in the trophic pyramid of the soil's microbial ecosystem, predation of bacteria, fungae, etc., by larger organisms continues. It may be this that has liberated microbial biomass P; with no forest root/mycorrhizal system to retrieve it. Leaching would have achieved the rest under 4200 mm of annual rainfall.

The method of Brookes et al. [38] involves shaking the soil samples with, and without, a P 'spike', in NaHCO$_3$, following fumigation of one set of samples. The 'spike' calibrates sorption. This, in a temperate soil, would not be expected to be very significant. In these acid ultisols (pH 4.1), high sorption levels distort the data which cannot usefully be extrapolated. I suggest that fumigation and subsequent extraction through leaching tubes, without shaking and dispersion, could yield interesting results here. Lack of time and equipment prevented this happening in the Cambridge Alley-cropping Projects in Costa Rica.

# 19. The importance of rock phosphate in sustaining the production of basic grains

On the San Juan site, the main experimental plots were replicated and randomized among four treatments:

—with, or without, trees in alley configuration;
—with, or without, the application of rock-P.

An application of 40 kg P ha$^{-1}$ was added to half of the plots, as rock-P, following the first (baseline) maize crop, and before the first bean crop. A further 40 kg P was added in December of year 2; and a further 20 kg one year later. Its effects were observed over the whole 7 years of the project. Table 4 gives an example from year 5 of the project.

**Table 4.** The ability of differing treatments to export, from the soil, the limiting nutrient in the form of grain: a useful measure of the system's efficiency. (San Juan site: year 5. P exports in grain: mg P m$^{-2}$ (s.e. $\pm$).)

|  | clear plots | | | | alley plots[a] | | | |
|---|---|---|---|---|---|---|---|---|
|  | −P | | +P | | −P | | +P | |
| beans | 189.6 | (29.1) | 374.4 | (36.9) | 211.5 | (7.4) | 412.0 | (38.5) |
| maize | 234.6 | (35.9) | 427.0 | (45.6) | 208.3 | (7.0) | 397.2 | (20.2) |
|  | 424.2 | (46.2) | 801.4 | (58.7) | 419.8 | (10.1) | 809.2 | (43.5) |

[a]Alley plots here were comprised of alternating rows of Gliricidia sepium and Erythrina fusca.

The effect is seen, not only, in higher grain yields, but also, in the higher P-content of the grain. The long-term effect of the rock-P applications can be seen in the Inga alley data for year 7 (table 5).

**Table 5.** The ability of differing treatments to export, from the soil, the limiting nutrient in the form of grain: a useful measure of the system's efficiency. (P exports in grain (maize + beans): year 7, San Juan site. Clear plots and *Inga edulis* alleys: kg P ha$^{-1}$.)

| treatment → | clear (control) plots | *I. edulis* alleys zero P | *I. edulis* alleys + P |
|---|---|---|---|
| P-exported in grain | 4.28 | 8.40 | 12.72 |
| s.e. | 0.53 | 0.39 | 0.82 |

Sustainability, in this context, required high inputs of organic matter to sustain the microbial population, together with adequate supplies of the key limiting nutrient in a form that does not leach and, thus, persists in the soil.

# 20. Responses to cation applications

**Table 6.** The effect of varying additions of a cation mixture upon grain yields; 4 years post-burn: San Juan site: year 4. (Cation micro-plots set within clear plots. Bean yields: grams per plant.)

| cation mixture applied: kg ha$^{-1}$ → | 0 | 200 | 400 | 600 | 800 |
|---|---|---|---|---|---|
| bean yield. clear plots: zero P | 10.7 ± 1.2 | 8.5 ± 1.2 | 12.1 ± 1.0 | 12.1 ± 2.0 | 12.2 ± 1.6 |
| bean yield. clear plots: + P | 17.8 ± 1.2 | 20.2 ± 1.0 | 19.7 ± 2.5 | 20.0 ± 2.1 | 20.9 ± 2.1 |

The cation mixture was 4 : 1 (w/w) ground dolomitic lime + 'K-Mag' ($K_2SO_4$ + $MgSO_4$).

The data in table 6 show no response in bean yields to the varying levels of the cation mixture, but do show a consistent response to the rock-P 4 years after the first application.

The implementing NGO has no numerical data for the markedly different response of developing *I. edulis* alleys on very degraded soils in northern Honduras, as described earlier. In applying these findings some years later, and in a different context, the cation mixture has proved itself indispensable and is now applied, mixed with the rock-P, at planting out.

# 21. The phosphorus paradox

The findings highlighted above regarding the role of P in food security on these acid soils throw up another conundrum: crop production on both the San Juan and Sarapiquí sites was limited by available P before any other factor; the P contained in the above-ground biomass had been completely leached through the soil profile shortly after the burn; so, the question arises: what is the source of the P that is clearly available to the successful first crops following a burn?

Hands *et al.* [24] argue that the key P resource is the SOM accumulated, in acid conditions, under the forest or fallow; and argue further that this SOM undergoes an enhanced decomposition resulting from the liming effect of the ash, together with soil-heating and repeated wetting-drying cycles [42]. The ash effect is manifest in the temporary rise in soil pH observed by all studies. The effect may endure for no more than, perhaps, a year, depending upon the base status of both soil and biomass [43]. The study of da Silva Neto *et al.* [17] on an Amazonian ultisol showed that base-saturation peaked at two to three months post-burn and dropped below its starting value within 12 months; no changes occurred below 20 cm depth. However, acid-extractable P in the top 10 cm only, halved from the forest value immediately post-burn and continued to fall away. I argue above that dispersion/extraction techniques can distort observed P-values in these soils.

Hands *et al.* [24] argue that the combined factors of pH-change and exposure acting on SOM liberate, in a 'pulse', the P, N and S contained in the more labile fractions of SOM. Labile soil organic P is a finite resource [42] and we suggest that it is this that explains the rapid loss of fertility within a short time of the

burning operation; and that it is the retrieval, retention and recycling of P from rock-P that is the key to a sustainable alternative.

Leaving this argument aside, the outcome of the long-term trials on the San Juan site was that Inga A-C, supplemented by rock-P, sustained the production of its own biomass and of yields in its basic grains over the whole 7-year project; and for longer in the later Honduras trials.

## 22. The wider context of agroforestry and soil properties

The detailed description, outlined earlier, of certain, apparently key, soil properties and their amelioration, fits within a broader and more-recent agroforestry picture; but I contend that it amends that picture. An example is given by the analysis of meta-data by Muchane *et al.* [44] who compiled data across a range of agroforestry practices in the humid and sub-humid tropics. They set four hypotheses within each of six questions relating to soil properties: (i) does agroforestry reduce soil erosion? (ii) does agroforestry build-up soil organic C stocks? (iii) does agroforestry build-up soil N stocks? (iv) does agroforestry increase availability of soil N? (v) does agroforestry increase availability of soil P? and (vi) does agroforestry ameliorate soil acidity?

The four agroforestry contexts were: (i) agroforestry in general; (ii) simultaneous agroforestry (e.g. alley-cropping) versus sequential agroforestry (e.g. managed fallows); (iii) agroforestry including N-fixing trees; and (iv) agroforestry in sandy soils.

The hypotheses were broadly upheld with high confidence within the first three questions and either supported with low confidence or unsupported by the data in the remaining three.

I would add that, as described earlier, deep mulching renders question (vi) (soil acidity) largely irrelevant in Inga A-C, as the rooting systems tend to be concentrated within very shallow soil layers; and that small additions of cations to these layers appear to obviate the problem in the present Land for Life Program. Also, that high inputs of mulch feed the soil microbiota and maintain higher P-availability than in bare-soil plots, as illustrated in basic grain P-content for year 7 post-burn in table 5. Moreover, because Inga A-C in the Land for Life Program sustains acceptably high yields of basic grains without N additions, we infer (but have not proved) adequate and sustained availability of soil N. Using published foliar N-contents for *I. oerstediana* and reliable mulch production data from the San Juan site [25], we estimated an annual cycling of over 250 kg ha$^{-1}$ N within the system.

## 23. Components of the Guama (Inga) Model

### 23.1. The enabling component of the Guama Model

#### 23.1.1. Inga alley-cropping for food security in basic grains

As described earlier, this is the key component that enables the family to establish other, permanent agroforestry systems on land that no longer needs to be held in reserve for future S-B operations.

### 23.2. Other components of the Guama Model

#### 23.2.1. Inga alley-cropping for cash crops

The best example is pepper (*Piper nigrum*) grown, within *Inga* alleys, on living stakes of *Gliricidia sepium* (figure 13). These are positioned 3 m apart in the centre of the alleys, but slightly zig-zagged each side of the centre-line. On the demonstration farm at Las Flores, we have interplanted turmeric (*Curcuma longa; syn. Curcuma domestica*). Both cultivars thrive in the high organic matter soil environment and are unrelated botanically.

Another high-value cultivar grown similarly on 'bridged' stakes of *G. sepium* is vanilla (*Vanilla planifolium*). Management of the *Inga* in these roles differs from that with the light-demanding basic grains; the *Inga* are pruned sporadically and lightly, at times that harmonize with the phenology of the cultivars. Consistent results at La Flores indicate that the income from these kinds of cash crops can transform the family's economy. It goes without saying that this is dependent on the world and local commodity prices, but the great advantage to the families is that these are minimal-input, debt-free systems; they depend upon biological, rather than economic, processes. Risks are low and margins are wide; giving flexibility and resilience against economic or climatic shock.

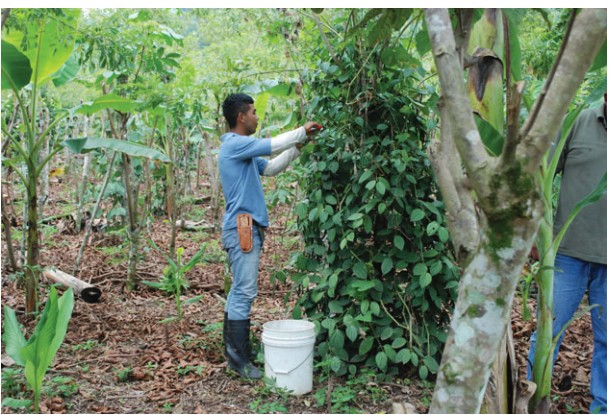

**Figure 13.** Las Flores. Pepper (*Piper nigrum*) on living stakes of *Gliricidia sepium* within *Inga edulis* alleys. The pepper is interplanted with developing turmeric (*Curcuma longa*) and plantains (*Musa* sp.). The same alley as figure 6.

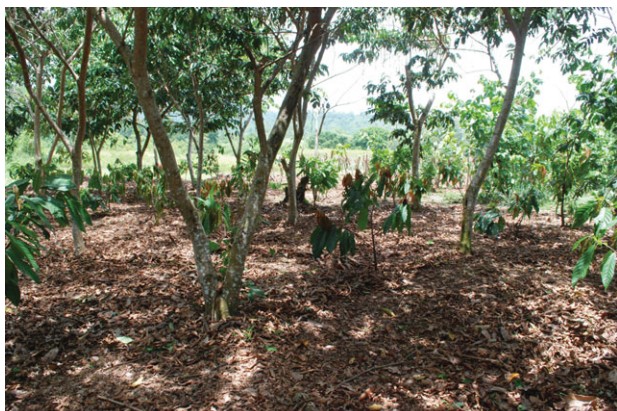

**Figure 14.** CURLA site. Young cacao developing beneath the shade of *Inga edulis*. Weed control here is largely achieved by shade. This site had been dominated by invasive grasses prior to its recapture by *Inga*.

### 23.2.2. *Inga* combined with fruit trees

The best example is of cacao grown beneath the shade of *Inga* which is its only source of N (figures 14 and 15).

Total tree density is about $600 \, ha^{-1}$; cacao : *Inga* at about $3 : 1$. To accommodate optimum light requirements of the cacao, the shade is controlled by sporadic lopping of *Inga* branches. This, in turn, can yield valuable domestic firewood in steady quantities throughout the year. Foliage from the branches is spread beneath the cacao as mulch. The present programme is seeing a high demand from the families for top-quality grafted or hybrid cacao which we donate as part of the strategy to establish a high concentration of families successfully implementing the Guama Model. Since 2012, over 300 000 cacao saplings have been distributed in this way. Once it becomes productive, cacao can provide a modest, but steady, income for perhaps 10–11 months of the year with no sudden peaks in labour demand.

A similar number of other fruiting trees of the following cultivars have been grafted and distributed: avocado; citrus; rambutan and, more recently, allspice. Our present strategy is to suggest that the families plant two staggered rows of the fruit trees next to two of the same of *Inga* in $2 : 2$ configuration across the site. The canopies intermesh; and any tendency for the *Inga* to overshade the fruit trees is controlled, as above, by sporadic lopping of branches. This is a provisional recommendation which we shall review and amend, if necessary. *Inga* trees, planted in this way, also function as a seed bank for future expansion of the Model.

The Land for Life demonstration farm at Las Flores in the Cuero valley has a land-use history of over 100 years of S-B. The *Inga* canopy over cacao has been markedly thin since planting and we have recently applied the mineral supplement mixture, with immediate success.

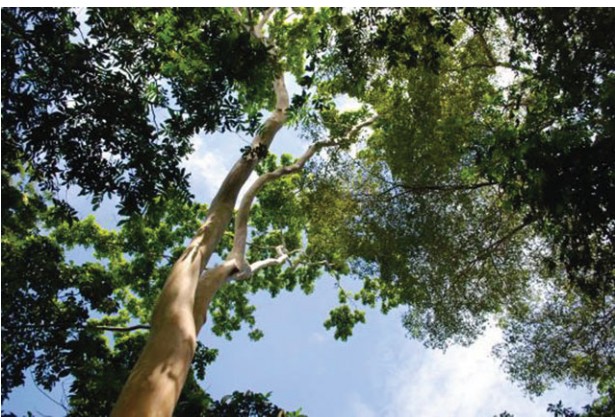

**Figure 15.** CURLA site: The Biological Corridor. 14-year-old *Terminalia oblonga* as an emergent from the Inga canopy. Left: *Inga vera*. Right: *Hymenaea courbaril* and *I. vera*.

A further refinement of the *Inga*-cacao model is to replace a few *Inga* with light-canopied timber trees of high value. The rosewoods (*Dalbergia* spp.) are obvious candidates as legumes. They tend to defoliate annually, at times that suit peak light-demand of the cacao, prior to re-flushing their foliage and flowering. The non-legume *Cedrela odorata* (Meliaceae) functions similarly and holds very high value as timber. Prices of these, and other, highly prized fine timbers have soared in recent years and show no sign of dropping.

### 23.2.3. *Inga* as a framework or 'matrix' species for high-value tropical timber

The saplings of many rain forest trees struggle when planted into invasive grassland. The *Inga* species under trial at San Juan and, later, at CURLA are all tolerant of acid soils and all are highly competitive. In 2000, the Cam project planted 2 ha of such grassland with *Inga* spp. at 4 m spacing and 'tresbolillo' (crowsfoot) configuration, in order to begin the process of site-recapture prior to inter-planting with 15 species of rain forest trees. One *Inga* in four was omitted to provide hexagonal gaps for the forest trees. The competitor grass species included the notoriously hard-to-control *Rottboellia cochinchinensis* and the unexpectedly tough *Digitaria swazilandensis*. The *Inga* needed a couple of passes with the machete to hold back the grasses while they established themselves, but, once up to 1–1.5 m, were able to grow on to dominate the grasses. Before complete canopy closure, the forest species saplings were planted into their narrowing gaps. Twenty years later, most of the latter now tower above their *Inga* 'Nodriza' (nurse) companions and the reforested strip is functioning as a biological corridor. The soil beneath it has been transformed; and in the understorey are trees (e.g. *Xylopia frutescens* (Annonaceae)) that can only have germinated from seed brought in by birds; mainly toucans, brown jays, etc. The canopy is currently at about 20–25 m; with emergents (*Terminalia* spp., *Swietenia* sp.) many metres higher (figure 16).

Figure 17 shows the theory regarding the combination of *Inga* spp. with fruit or timber trees. Since 2012, the project has distributed around 500 000 fruit tree, and about 100 000 timber, saplings. However, I have to report that the reality of the family plots is much more ragged than these neat ideas imply. Many timber trees are planted by them around the borders of the holding or in sporadic clusters with, or without, *Inga* as companion trees.

## 24. Agroforestry and atmospheric carbon

Based on the Guama Model, a derived Carbon Model assumes an average family holding of about 8 ha; it assumes the family's planting of 2 ha of Inga A-C and ceasing to S-B. It assumes modest *Inga*/timber plantings of 1 ha per year by the family from year 3 following their adoption. The Carbon Model estimates that, by the end of year 12, the average family's annual rate of Carbon (C)-sequestration and avoidance will be 146 tonnes of C (and rising); their accumulated total C will be 1166 t C (or 4314 t $CO_2$) by that date.

Recruiting at 40 families per year since 2012, the Land for Life Program is thus estimated to have sequestered/avoided 284 000 t $CO_2$ by the end of 2019.

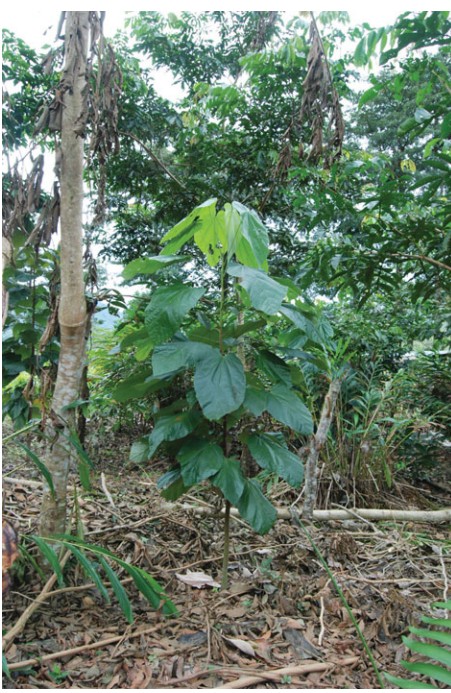

**Figure 16.** Las Flores. February 2018. Young *Theobroma bicolor* within the *Inga* matrix and among clumps of cardamom (*Elettaria cardamomum*). This species shows dimorphism in the leaves. In the case of stem-leaves, the upper cluster rises above the terminal bud in the form of a 'parasol'. Branch leaves are simple, alternate and much smaller. This species is thought to have been used by the ancient Maya. The fermented seeds were mixed with cured vanilla, ground and mixed with honey to make the original 'Xocolatl'. When *T. cacao* was brought from the Amazon, it was preferred.

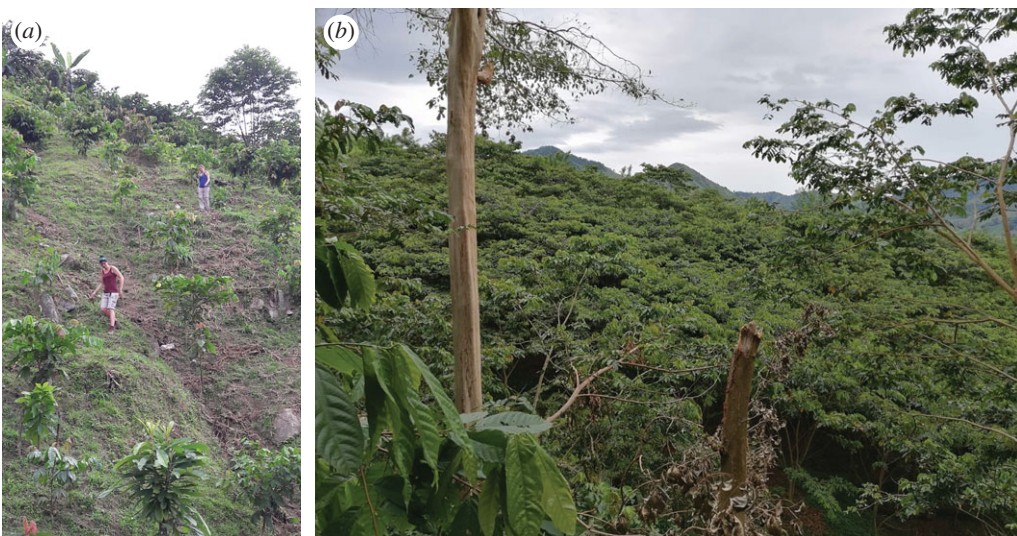

**Figure 17.** (*a*) Reforestation of degraded pasture. Las Flores, San Marcos, Cuero valley, March 2017. Newly purchased land adjacent to the Inga Foundation demonstration farm at Las Flores. *Inga edulis* and *I. oerstediana* planted as 'matrix' or framework species; later to be interplanted with cacao and a wide mixture of rain forest trees. (*b*) Reforestation of degraded pasture: the arboretum at Las Flores. The same slope in May 2020. Seen from a distance, it resembles a low-stature forest. It functions as an agroforest and arboretum for rare and endangered tree species. In time, the interplanted trees will overtop the *Inga* spp., branches of which will be lopped sporadically to allow light onto the cacao. A spring has started flowing close to the location of the lower lady in (*a*).

## 25. Landscape regeneration

The *Inga*/fruit trees and *Inga*/timber tree associations outlined here, develop rapidly and come to resemble a secondary forest. These plantings abound with birds, reptiles, mammals and insects. Our

colleagues in Panthera (Honduras) (F. Castañeda 2018, personal communication) have taken camera-trap photographs of pumas, ocelots and jaguars within such cacao plantings.

We have consistent testimony from Inga farmers, and recent evidence from our own farm, that these agroforestry systems conserve soil moisture; springs start running where previously there had only been occasional damp ground.

The implementing NGO and a local partner (MOPAWI) were asked, in 2019, by IUCN and the Honduran Ministry of Natural Resources and Environment (SERNA-MiAmbiente) to submit a paper on the *restoration of degraded former rain forest landscapes*. We were able to make the point that basic human need had driven the original deforestation and that people must be the key factors in restoration. Moreover, we argued that the opposite of deforestation is not necessarily simple 'reforestation'. Restoration of the landscape has to imply the reversal of soil degradation; the restoration of SOM and the restoration of the microbial populations that formerly retrieved, retained and recycled essential nutrients, most of which have been lost and will have to be replenished. The combination of trees and mineral supplements achieves this.

# 26. Conclusion

The role of P in rain forests and in humid tropical subsistence farming is central to this paper and to the functioning of the Guama Model described earlier. Until recent decades, the literature on the role of P in S-B farming on highly weathered latosols had been inconclusive and, in some instances, contradictory. Until this web of unclarity could be untangled, progress towards a sustainable alternative remained hindered. The confusion prevailed in a form of 'double bind'; thus:

(i) P might, or might not, be the key limiting nutrient in the functioning of primary or secondary rain forests, and subsequent subsistence agriculture, on the highly weathered and leached latosols that are typical of humid tropical environments; and

(ii) These latosols 'fix' (sorp) both inorganic and soluble organic forms of P. Therefore, P is not leached through the soil profile but is held by the reactive sesquioxide clays. (Perhaps, 'banding' (concentrating) P fertilizers along crop rows might work [11].)

Considerable space in this paper has been occupied to show that these soils do leach P; and do so readily via preferential flow micropores. The confusion resulting from the two assumptions outlined above stems from the fact that these soils exhibit radically different behaviour *in vitro* in the soil laboratory than they do *in situ*.

Moreover, attempts to demonstrate that cropping systems, or the forest itself, are, or are not, P-limited added to the confusion by using readily soluble forms of P fertilizer. It is one contention of this paper that lack of response in the crops or vegetation is owing to the rapid leaching of P applied in this form and is not owing to the absence of the need for P.

The application of rock-P to half of the experimental plots in the Cambridge Alley-cropping Projects provoked an immediate and definitive response in both the basic grain crops and in the alley trees themselves. This response was observed over the entire trial period of 7 years. Alley-cropping with *Inga* spp., supplemented by rock-P emerged from 7 years' trial as the only system, of those trialled, promising sustainability in this context.

The maintenance of organic matter inputs to feed the soil microbial population, supplemented by P as rock-P proved itself to be the basis of the sustainable production of basic grains; and thus an alternative to shifting agriculture. On highly degraded humid tropical soils, supplementary sources of Ca, K and Mg cations are also necessary to assist the agroforest vegetation in the recapture of the site from invasive grasses and to retrieve, retain and recycle essential nutrients.

Once food security in basic grains can be achieved with Inga A-C, supplemented in this way, the family can abandon S-B and plant their remaining land with productive trees.

Re-greening with agroforest will be neither cheap, nor quick, nor easy. It will, however, be effective, permanent and resilient to climatic violence. An integrated agroforest strategy can lift subsistence farming families out of poverty and food insecurity; it can re-invigorate the rural economy; it can restore catchment protection and habitat; and it can halt further burning. Massive $CO_2$ emissions are thus avoided and immense volumes of atmospheric $CO_2$ can be sequestered into soil and vegetation; in the latter, for decades to come.

Ethics. No unethical practices occurred regarding people or animals during these studies. The Cambridge Projects' activities on the San Juan and Sarapiquí sites, and later in Honduras, took place with the full knowledge, cooperation and consent of the respective landowners.

Data accessibility. Most of the data cited are already published. They are included in the text as, and where, they are relevant to the narrative. Original hand-written field and laboratory records from the Cambridge Alley-cropping Projects are held at Higher Penhale, Lostwithiel, Cornwall PL22 0HY, UK. They are very bulky and none have yet been stored digitally.

Competing interests. The author has no competing interests.

Funding. 1986–1988. The Worts Travelling Scholars' Fund, University of Cambridge; The Philip Lake Fund, University of Cambridge; The Royal Society Dudley Stamp and 20th International Geographical Congress Funds, Wolfson College. Cambridge, 1988–1994. Commission of the European Communities. DG XII: Science and Technology for Development. Projects TS2 0029 and TS3-CT910021: Nutrient Cycling and Sustainability in Alley-cropping systems in the Humid Tropics: I and II. Department of Geography, University of Cambridge. 1994–2002. Commission of the European Communities. DG 1: North-South Relations. Tropical Forests Budgetary Line Project B7-5041/1/94/07: Alley-Cropping as a Sustainable Alternative to Shifting Cultivation: III. Project HND/B7-6201/IB/97/0533(08): Alley-Cropping as a Sustainable Alternative to Shifting Cultivation: IV. Department of Geography, University of Cambridge. 2007–present. The Inga Foundation. UK-registered charity: 1124688. Private donations and grants from charitable institutions.

Acknowledgements. The help and support of Professor Timothy Bayliss-Smith, expressed over the course of many years during these studies and projects, is gratefully acknowledged. In Costa Rica, a long collaboration with the Centro de Investigaciones Agricolas of the University of Costa Rica made the Cambridge studies possible there. I am most grateful to Dr Freddy Sancho, Dr Alfredo Alvarado, Dr Gabriela Soto and many others for the pleasure of their collaboration. I am grateful for the contributions of many graduate and other volunteers in generating much of the field and laboratory data, mostly published elsewhere and used here to illustrate points in the narrative. In Honduras, fruitful collaborations continue after many years with the Centro Universitario Regional del Litoral Atlántico (CURLA) of the Universidad Autónoma de Honduras (UNAH); and, for over 25 years, with Mosquitia Pawisa (MOPAWI). Ing. Abraham Martinez and Ing. Luis Miranda are outstanding exponents of the Guama Model in Honduras. They have taught many others these techniques and much of this paper's later content describes their insights and successes. I am grateful for the most generous help of Dr David Guiterman in carrying out a thorough review of the scientific and grammatical English in this paper. Any residual errors are entirely my own responsibility.

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
