## [Peer Review File · Royal Society Open Science]

Review History

RSOS-201204.R0 (Original submission)

Review form: Reviewer 1

Is the manuscript scientifically sound in its present form?

Yes

Are the interpretations and conclusions justified by the results?

Yes

Is the language acceptable?

No

Do you have any ethical concerns with this paper?

No

Have you any concerns about statistical analyses in this paper?

Yes

Recommendation?

Reject

Comments to the Author(s)

While you have very interesting and important information hidden away in this paper, it sadly is not discussed comparatively with many other similar studies implemented in other parts of the world since 2000. The paper also contains far too much irrelevant information about the history of your Programme. If the Editors do accept your paper it needs to be completely rewritten in a different style and brought up-to-date.

Review form: Reviewer 2

Is the manuscript scientifically sound in its present form?

No

Are the interpretations and conclusions justified by the results?

Yes

Is the language acceptable?

Yes

Do you have any ethical concerns with this paper?

No

Have you any concerns about statistical analyses in this paper?

No

Recommendation?

Accept with minor revision (please list in comments)

Comments to the Author(s)

This manuscript (RSOS 201204) deals with 'The search for a sustainable alternative to slash-and-burn agriculture in the World's Rain Forests: The Guama Model and Inga Foundation's Land for Life Program', and its main goals are 'achieving food-security in basic grains, on minimal inputs, and of providing the means of eliminating further slash-and-burn in the region'. It is about a long term study of the alley cropping systems using Inga trees and conducted in Central America. It is a descriptive work, it cites many results conducted over the years by the researcher and his team. It is of a transdisciplinary nature, since it includes areas regarding food security, slash and burn agriculture, soil degradation, biodiversity, among others. The paper can be helpful for the international community working on sustainable food production, land degradation and soil health in rainforest regions, particularly at this stage when the world is approaching the planetary boundaries. However, the paper needs some minor corrections cited below.

Specific comments:

Abstract

The last paragraph it would be better if it is integrated within the first paragraph. So, that way, the conclusions which is about the Inga alley cropping benefits will stand out at the last lines of the section.

Page 3 (Introduction): since the paper is about agriculture, some mentions of agroecology would be appropriate, for example, citing the UN report on the Right to Food.

Page 5: lines 7-14, add more recent references about slash and burn agriculture. Line 34, add more recent references about phosphorus or specify that, that was the hypotheses at the time when the study began.

Page 6: lines 2-4, clarify the difference between the 'usual p.p.m' and ppm, are they the same?. Lines 33-34, was a readily-soluble P fertilizer applied to some of the plots?. Line 46, I think there is a typo 'reaon' instead of reason?.

Page 7: line 22, is it affect? instead of effect?. Line 26, integrated rural livelihood model. Line 44, are the use of quotations in "guama" necessary?. Line 45, is there a source about the 8 ha average farm size that can be cited?, probably it is less. Line 52, 5000 inga trees/ha?

Page 8: line 20, does CA stand for Central America?

Page 9: line 9, explain how dolomites and K-Mg mixture works, if possible, provide some insights into the interactions among nutrients. Could have the responses been achieved only with dolomite? Since it is a common practice for farmers to use it in acid soils, and would be cheaper for them, if only dolomite is used.

Page 10: lines 1 - 7, clarify the interaction between ants and wasps vs. pests (e.g. do they parasitize the larvae?).

Page 11: line 39, is it °C, instead of C?

Page 14: Formatting of Table 6 (adding lines).

Page 15: line 61, Centro Universitario Regional del Litoral Atlántico.

General comments:

Throughout the manuscript, the names of elements (e.g. Phosphorus, etc.), names of crops should not be capitalized. Also, avoid dots after SI units (e.g. kg., m., ha., etc.).

Add a conclusion section, it would be helpful to summarize the main findings, even though some conclusions are provided, but putting them in a conclusion section would be good for clarity.

Add more recent references, whenever possible. For example, about the P paradox, consider: Turner, B., Brenes-Arguedas, T. & Condit, R. 2018. Pervasive phosphorus limitation of tree species but not communities in tropical forests. *Nature* 555, 367–370. <https://doi.org/10.1038/nature25789>

Decision letter (RSOS-201204.R0)

Dear Mr Hands

The Editors assigned to your paper RSOS-201204 "The search for a sustainable alternative to slash-and-burn in the World's Rain Forests: The Guama Model and Inga Foundation's Land for Life Program." have now received comments from reviewers and would like you to revise the paper in accordance with the reviewer comments and any comments from the Editors. Please note this decision does not guarantee eventual acceptance.

Please submit your revised manuscript and required files (see below) no later than 21 days from today's (ie 21-Aug-2020) date. Note: the ScholarOne system will 'lock' if submission of the revision is attempted 21 or more days after the deadline. If you do not think you will be able to meet this deadline please contact the editorial office immediately.

on behalf of Dr Agnieszka Latawiec (Associate Editor) and Pete Smith (Subject Editor)
openscience@royalsociety.org

Reviewer comments to Author:
Reviewer: 1

Comments to the Author(s)

While you have very interesting and important information hidden away in this paper, it sadly is not discussed comparatively with many other similar studies implemented in other parts of the world since 2000. The paper also contains far too much irrelevant information about the history of your Programme. If the Editors do accept your paper it needs to be completely rewritten in a different style and brought up-to-date.

Reviewer: 2

Comments to the Author(s)

This manuscript (RSOS 201204) deals with 'The search for a sustainable alternative to slash-and-burn agriculture in the World's Rain Forests: The Guama Model and Inga Foundation's Land for Life Program', and its main goals are 'achieving food-security in basic grains, on minimal inputs, and of providing the means of eliminating further slash-and-burn in the region'. It is about a long term study of the alley cropping systems using Inga trees and conducted in Central America. It is a descriptive work, it cites many results conducted over the years by the researcher and his team. It is of a transdisciplinary nature, since it includes areas regarding food security, slash and burn agriculture, soil degradation, biodiversity, among others. The paper can be helpful for the international community working on sustainable food production, land degradation and soil health in rainforest regions, particularly at this stage when the world is approaching the planetary boundaries. However, the paper needs some minor corrections cited below.

Specific comments:

Abstract

The last paragraph it would be better if it is integrated within the first paragraph. So, that way, the conclusions which is about the Inga alley cropping benefits will stand out at the last lines of the section.

Page 3 (Introduction): since the paper is about agriculture, some mentions of agroecology would be appropriate, for example, citing the UN report on the Right to Food.

Page 5: lines 7-14, add more recent references about slash and burn agriculture. Line 34, add more recent references about phosphorus or specify that, that was the hypotheses at the time when the study began.

Page 6: lines 2-4, clarify the difference between the 'usual p.p.m' and ppm, are they the same?. Lines 33-34, was a readily-soluble P fertilizer applied to some of the plots?. Line 46, I think there is a typo 'reanon' instead of reason?.

Page 7: line 22, is it affect? instead of effect?. Line 26, integrated rural livelihood model. Line 44, are the use of quotations in "guama" necessary?. Line 45, is there a source about the 8 ha average farm size that can be cited?, probably it is less. Line 52, 5000 inga trees/ha?

Page 8: line 20, does CA stand for Central America?

Page 9: line 9, explain how dolomites and K-Mg mixture works, if possible, provide some insights into the interactions among nutrients. Could have the responses been achieved only with dolomite? Since it is a common practice for farmers to use it in acid soils, and would be cheaper for them, if only dolomite is used.

Page 10: lines 1 - 7, clarify the interaction between ants and wasps vs. pests (e.g. do they parasitize the larvae?).

Page 11: line 39, is it °C, instead of C?

Page 14: Formatting of Table 6 (adding lines).

Page 15: line 61, Centro Universitario Regional del Litoral Atlántico.

General comments:

Throughout the manuscript, the names of elements (e.g. Phosphorus, etc.), names of crops should not be capitalized. Also, avoid dots after SI units (e.g. kg., m., ha., etc.).

Add a conclusion section, it would be helpful to summarize the main findings, even though some conclusions are provided, but putting them in a conclusion section would be good for clarity.

Add more recent references, whenever possible. For example, about the P paradox, consider: Turner, B., Brenes-Arguedas, T. & Condit, R. 2018. Pervasive phosphorus limitation of tree species but not communities in tropical forests. *Nature* 555, 367–370.
<https://doi.org/10.1038/nature25789>

===PREPARING YOUR MANUSCRIPT===

- one version identifying all the changes that have been made (for instance, in coloured highlight, in bold text, or tracked changes);
- a 'clean' version of the new manuscript that incorporates the changes made, but does not highlight them. This version will be used for typesetting if your manuscript is accepted.

===PREPARING YOUR REVISION IN SCHOLARONE===

Author's Response to Decision Letter for (RSOS-201204.R0)

See Appendix A.

RSOS-201204.R1 (Revision)

Review form: Reviewer 1

Is the manuscript scientifically sound in its present form?

Yes

Are the interpretations and conclusions justified by the results?

Yes

Is the language acceptable?

No

Do you have any ethical concerns with this paper?

No

Have you any concerns about statistical analyses in this paper?

No

Recommendation?

Accept with minor revision (please list in comments)

Comments to the Author(s)

Dear Mike

Your paper is solid and presents interesting and important content. Please check with the Editors if the form is accepted by the journal as the paper does not strictly follow the order of intro, methods, results etc. I would also remove the details of your Institute.

Review form: Reviewer 2

Is the manuscript scientifically sound in its present form?

Yes

Are the interpretations and conclusions justified by the results?

Yes

Is the language acceptable?

Yes

Do you have any ethical concerns with this paper?

No

Have you any concerns about statistical analyses in this paper?

No

Recommendation?

Accept as is

Comments to the Author(s)

This manuscript (RSOS 201204) entitled 'The search for a sustainable alternative to slash-and-burn in the World's Rain Forests: The Guama Model and Inga Foundation's Land for Life Program' has been improved compared to the first manuscript. It is an interesting study and the author has summarized a long term research, combining field trials and laboratory work. The paper (if accepted) would be useful as input for policy making regarding the land use in the humid tropical regions, and also for education in the field of agronomy and forestry.

My suggestions:

The format of naming of Tables need to be improved, make it in a more descriptive manner.

In p. 17, line 10, it is tresbolillo, instead of trebolillo.

Decision letter (RSOS-201204.R1)

Dear Mr Hands

On behalf of the Editors, we are pleased to inform you that your Manuscript RSOS-201204.R1 "The search for a sustainable alternative to slash-and-burn in the World's Rain Forests: The Guama Model and Inga Foundation's Land for Life Program." has been accepted for publication in Royal Society Open Science subject to minor revision in accordance with the referees' reports. Please find the referees' comments along with any feedback from the Editors below my signature.

Please submit your revised manuscript and required files (see below) no later than 7 days from today's (ie 21-Oct-2020) date. Note: the ScholarOne system will 'lock' if submission of the revision is attempted 7 or more days after the deadline. If you do not think you will be able to meet this deadline please contact the editorial office immediately.

on behalf of Dr Agnieszka Latawiec (Associate Editor) and Pete Smith (Subject Editor)
 openscience@royalsociety.org

Associate Editor Comments to Author (Dr Agnieszka Latawiec):

Although the author is a native speaker I think the manuscript could be improved with respect to use of academic/scientific English. I therefore suggest the manuscript to be revised by professional scientific English editing service before the manuscript is accepted.

The information about the establishment and the history of the Inga Institute is not appropriate for the paper. It can be acknowledged given financing of initiatives but in the text it may appear as an advertisement. I would also remove it from the title.

Reviewer comments to Author:

Reviewer: 1

Comments to the Author(s)

Dear Mike

Your paper is solid and presents interesting and important content. Please check with the Editors if the form is accepted by the journal as the paper does not strictly follow the order of intro, methods, results etc. I would also remove the details of your Institute.

Reviewer: 2

Comments to the Author(s)

This manuscript (RSOS 201204) entitled 'The search for a sustainable alternative to slash-and-burn in the World's Rain Forests: The Guama Model and Inga Foundation's Land for Life Program' has been improved compared to the first manuscript. It is an interesting study and the author has summarized a long term research, combining field trials and laboratory work. The paper (if accepted) would be useful as input for policy making regarding the land use in the humid tropical regions, and also for education in the field of agronomy and forestry.

My suggestions:

The format of naming of Tables need to be improved, make it in a more descriptive manner. In p. 17, line 10, it is tresbolillo, instead of trebolillo.

===PREPARING YOUR MANUSCRIPT===

Your revised paper should include the changes requested by the referees and Editors of your manuscript. You should provide two versions of this manuscript and both versions must be provided in an editable format:
 one version identifying all the changes that have been made (for instance, in coloured highlight, in bold text, or tracked changes);
 a 'clean' version of the new manuscript that incorporates the changes made, but does not highlight them. This version will be used for typesetting.
 Please ensure that any equations included in the paper are editable text and not embedded images.

Please ensure that you include an acknowledgements' section before your reference list/bibliography. This should acknowledge anyone who assisted with your work, but does not

qualify as an author per the guidelines at <https://royalsociety.org/journals/ethics-policies/openness/>.

===PREPARING YOUR REVISION IN SCHOLARONE===

- Ensure that your data access statement meets the requirements at <https://royalsociety.org/journals/authors/author-guidelines/#data>. You should ensure that you cite the dataset in your reference list. If you have deposited data etc in the Dryad repository, please only include the 'For publication' link at this stage. You should remove the 'For review' link.
- If you are requesting an article processing charge waiver, you must select the relevant waiver option (if requesting a discretionary waiver, the form should have been uploaded at Step 3 'File upload' above).
- If you have uploaded ESM files, please ensure you follow the guidance at <https://royalsociety.org/journals/authors/author-guidelines/#supplementary-material> to include a suitable title and informative caption. An example of appropriate titling and captioning may be found at [https://figshare.com/articles/Table_S2_from_Is_there_a_trade-off_between_peak_performance_and_performance_breadth_across_temperatures_for_aerobic_sc ope_in_teleost_fishes_/3843624](https://figshare.com/articles/Table_S2_from_Is_there_a_trade-off_between_peak_performance_and_performance_breadth_across_temperatures_for_aerobic_scope_in_teleost_fishes_/3843624).

Author's Response to Decision Letter for (RSOS-201204.R1)

See Appendix B.

RSOS-201204.R2 (Revision)

Review form: Reviewer 1

Is the manuscript scientifically sound in its present form?

Yes

Are the interpretations and conclusions justified by the results?

Yes

Is the language acceptable?

Yes

Do you have any ethical concerns with this paper?

No

Have you any concerns about statistical analyses in this paper?

No

Recommendation?

Accept as is

Comments to the Author(s)

While recommending acceptance, I do not think this is a strong paper and it would have been greatly improved by references to the agroforestry literature from other parts of the tropics which have moved on from alley cropping

Review form: Reviewer 2**Is the manuscript scientifically sound in its present form?**

Yes

Are the interpretations and conclusions justified by the results?

Yes

Is the language acceptable?

Yes

Do you have any ethical concerns with this paper?

No

Have you any concerns about statistical analyses in this paper?

No

Recommendation?

Accept with minor revision (please list in comments)

Comments to the Author(s)

The paper has been improved as compared to the previous version. However a few minor corrections can still be made: (below page numbers refer to pages in the paper, not the whole document).

1. In page 7 line 8, there is still mention to IF.
2. After the symbol of element phosphorus, the dot can be removed (e.g. page11 line 52, p 11 line 30, etc.).
3. The first time the mention to CURLA appears is in page 8, line 26, there its meaning must be provided, and conversely in page 16 line 4, only its acronym will suffice.

Decision letter (RSOS-201204.R2)

This year has been very difficult for everyone, and we want to take the opportunity to thank you for your continued support in 2020.

The Royal Society Open Science editorial office will be closed from the evening of Friday 18 December 2020 until Monday 4 January 2021. We will not be responding during this time. If you have received a deadline within this time period, please contact us as soon as possible to allow us to extend the deadline. If you receive any automated messages during this time asking you to meet a deadline, we offer apologies and invite you to respond after the festive period or during normal working hours.

With our best for a peaceful festive period and New Year, and we look forward to working with you in 2021.

Dear Mr Hands

On behalf of the Editors, we are pleased to inform you that your Manuscript RSOS-201204.R2 "The search for a sustainable alternative to slash-and-burn in the World's Rain Forests: The Guama Model and its Implementation." has been accepted for publication in Royal Society Open Science subject to minor revision in accordance with the referees' reports. Please find the referees' comments along with any feedback from the Editors below my signature.

Please submit your revised manuscript and required files (see below) no later than 7 days from today's (ie 16-Dec-2020) date. Note: the ScholarOne system will 'lock' if submission of the revision is attempted 7 or more days after the deadline. If you do not think you will be able to meet this deadline please contact the editorial office immediately.

on behalf of Dr Agnieszka Latawiec (Associate Editor) and Pete Smith (Subject Editor)
openscience@royalsociety.org

Associate Editor Comments to Author (Dr Agnieszka Latawiec):

Associate Editor: 1

Comments to the Author:

Please incorporate the suggestions of Reviewer 1.

Reviewer comments to Author:

Reviewer: 2

Comments to the Author(s)

The paper has been improved as compared to the previous version. However a few minor corrections can still be made: (below page numbers refer to pages in the paper, not the whole document).

1. In page 7 line 8, there is still mention to IF.
2. After the symbol of element phosphorus, the dot can be removed (e.g. page 11 line 52, p 11 line 30, etc.).
3. The first time the mention to CURLA appears is in page 8, line 26, there its meaning must be provided, and conversely in page 16 line 4, only its acronym will suffice.

Reviewer: 1

Comments to the Author(s)

While recommending acceptance, I do not think this is a strong paper and it would have been greatly improved by references to the agroforestry literature from other parts of the tropics which have moved on from alley cropping

===PREPARING YOUR MANUSCRIPT===

===PREPARING YOUR REVISION IN SCHOLARONE===

Author's Response to Decision Letter for (RSOS-201204.R2)

See Appendix C.

Decision letter (RSOS-201204.R3)

Dear Mr Hands,

It is a pleasure to accept your manuscript entitled "The search for a sustainable alternative to slash-and-burn in the World's Rain Forests: The Guama Model and its Implementation." in its current form for publication in Royal Society Open Science.

Best regards,

on behalf of Dr Agnieszka Latawiec (Associate Editor) and Pete Smith (Subject Editor)
openscience@royalsociety.org

Appendix A

RSOS-201204

The Search for Sustainable Alternative to slash-and-burn in the World's Rain Forests: The Guama Model and Inga Foundation's Land for Life Program

Detailed responses to reviewers' comments:

R 1's comments are generalised and dismissive. He/she requires that the subject matter be discussed in comparison with: "... similar studies implemented in other parts of the world since 2000." R 1 gives no examples. The context is slash-and-burn agriculture on the weathered and leached latosols that are typical of Rain Forest environments. I make the point that, although there is broad agreement across all studies regarding the impact of a slash-and-burn operation on soil cations, base status and Cation Exchange Capacity (CEC), impacts on soil phosphorus (P) remain elusive and, in some cases, contradictory. The fact is that no study since 2000 gives any data or insights into slash-and-burn agriculture that are fundamentally new.

In response to R1's comments I have cited the following: Cairns (2007) & (2014); da Silva Neto *et al* (2019); FAO (2014); FAO-CFS (2017); Leblanc *et al* (2006); McGrath *et al* (2001); & Pollini (2014).

Dr Latawiec has commented on my revised script:

"... The narrative indeed may be slightly different than some scientific works but if that is fine for the other RSOS Editors I am happy with it. The paper is extremely informative and contains important information."

The title of the RSOS Special Collection is: "Sustainable Land-use: Successful Initiatives ... etc". It invites descriptive and/or narrative accounts.

I have tried to show that confusion and contradiction prevailing in the literature in the mid 1980s had hindered the search for a sustainable alternative to slash-and-burn in this context; and that, moreover, the role of soil P is vital. No paper since 2000, with the possible exception of that of Turner *et al* (2018) (see below re. R 2's comments), throws any light on the role of soil P in this context. R1 contributes no relevant suggestion. I go on to argue that my experience in the laboratory with dispersion/extraction techniques is that they distort the true picture of soil P in a living, intact soil; a further layer of confusion regarding the vital role of soil P.

If the other Editors require that the narrative be reduced (e.g. re. the history of Inga Foundation's program), I am happy to comply; especially if they are specific regarding where they wish to see the cuts. The paper will not be: "... completely rewritten in a different style and brought up-to-date" as R1 so dismissively demands. It has been "brought up-to-date", insofar as it can be, in the absence of any new and significant insights.

Reviewer 2's comments are detailed and most helpful:

".... It is a descriptive work, it cites many results conducted over the years by the researcher and his team. It is of a transdisciplinary nature, since it includes areas regarding food security, slash and burn agriculture, soil degradation, biodiversity, among others. The paper can be helpful for the international community working on sustainable food production, land degradation and soil health in rainforest regions, particularly at this stage when the world is approaching the planetary boundaries. However, the paper needs some minor corrections cited below."

Taking R2's comments in his/her order, my responses are listed below:

Line numbers refer to those in the revised and uploaded manuscript:

- 1) Changes to the first and last paragraphs p.3;
- 2) Introduction: mention of Agroecology p.4 line 15 et sec.
- 3) More references, etc. p.4; line 17 et sec. p. 5 line 42, et sec.
- 4) More references to soil P p.5; line 60; p7; line 37
- 5) p.p.m or ppm all corrected to ppm
- 6) Was readily-soluble P added to some plots? p. 6; line 59; p.7; line 15
- 7) Typo "reaon" corrected
- 8) Various corrections to the original p. 7 all corrected.
- 9) A source re. 8 ha average holding p.8 line 32
- 10) CA stands for Central America addressed
- 11) Dolomite and K-Mag p. 10, line 34
- 12) Interaction between ants and wasps p. 10, line 57 et sec.
- 13) Re. pp 11, 14 15 of the original script all clarified and/or corrected.
- 14) names of elements, crops, SI units, etc all points corrected
- 15) Conclusion added as suggested. p.18
- 16) More recent references pp 4, 5, 6, 15.

I think that this addresses all those Reviewer comments that reasonably can be addressed.

Mike Hands
September 28th 2020

Appendix B

RSOS-201204

The Search for a Sustainable Alternative to slash-and-burn in the World's Rain Forests: The Guama Model and its implementation.

Detailed responses to Reviewers' and Editor's comments – 2nd Round: Responses to Dr. Guiterman's review of the English, etc.. Late October/early November 2020:

Associate Editor's Comments

- 1) References within the text to Inga Foundation, IF, etc. All references removed and the surrounding text amended to make sense.
- 2) Revision of academic/scientific English Dr. David Guiterman (Colleague) is (29th Oct 2020) revising the text. E-mails to RSOS on the 26th and 27th October refer; and include a request for a short time extension to enable the English revision.

Reviewer 1's comments

- 3) Re. the form of the paper Checked, as recommended by R 1, with Andrew Dunn (email, 26th October). This text is descriptive of a land-use system, its evolution and context; and does not take the form of a standard scientific paper.
- 4) Re. removal of references to Inga Foundation, IF, etc. All removed and text amended, as above.

Reviewer 2's comments

- 5) Re. the formatting of tables, titling of tables All table titles made self-explanatory, as recommended.
- 6) Re. p. 17 line 10 of the 2nd draft "Trebolillo" corrected to "Tresbolillo". I thank R 2 for correcting a long-held error. He/she is quite correct.

I thank both Reviewers and the Associate Editor for their most useful comments.

I thank Dr. David Guiterman for revising the English, the sense, the punctuation, etc. in this paper. His comments, requests for clarification and corrections have been incorporated in this latest, and, I hope, final, version.

David. Is this a correct statement?

I think that this addresses all points raised in the second round of the Review process.

Mike Hands
November 4th 2020

Appendix C

RSOS-201204: The Search for a Sustainable Alternative to slash-and-burn in the World's Rain Forests: The Guama Model and its implementation.

Detailed responses to Reviewers' comments – 3rd Round: December 2020:

Reviewer 1's comments appear to vacillate between the 2nd and 3rd reviews:

R 1's 2nd round comments, as reported to me by the Editors: October 2020 (sic):

"Your paper is solid and presents interesting and important content. Please check with the Editors if the form is accepted by the journal...". Points addressed in November.

3rd round comments:

"While recommending acceptance, I do not think this is a strong paper and it would have been greatly improved by references to the agroforestry literature from other parts of the tropics which have moved on from alley cropping"

This comment, requiring a wider field-of-reference, comes late in the review and revision process. No example is given in which alley-cropping is supposed to have "moved on" in this context. Interest in agroforestry in general has indeed widened in recent years; and I have referred to this in the paper. However, R 1 does not explain in what ways a system which has now proved itself as sustainable in humid tropical subsistence agriculture for over 25 years can be shown to have "moved on" in other tropical contexts; or in what ways it can be "moved on". I have taken pains to show that the cation nutrition of the Inga a-c system was modified in the light of recent field experience (pp. 7-8); and also that the generalised agroforestry literature misses important detail that takes alley-cropping out of the assumptions that had surrounded the system, as first conceived in West Africa. Perhaps other agroforestry systems need deeper scrutiny. What needs to move on, in my considered view, is political decision-making and the funding of re-greening initiatives.

The RS Special Collection calls for descriptions of: "... Successful Initiatives". This paper describes one such initiative, together with the ecological and scientific context in which it was developed: The acid soils that are typical of the humid tropics. It does not purport to be a review of agroforestry in general. However, in response to R 1's comment, I include reference to, and my comments on, a recent analysis of meta-data relating to agroforestry and soil properties in the humid and sub-humid tropics (Muchane, *et al.* 2020). Many more recent references can be found in the paper. They add breadth to the picture, but little that is fundamentally new to our understanding.

Reviewer 2's comments

"The paper has been improved as compared to the previous version. However a few minor corrections can still be made: (below page numbers refer to pages in the paper, not the whole document).

- 1. In page 7 line 8, there is still mention to IF.* removed; and surrounding text amended to make sense.
- 2. After the symbol of element phosphorus, the dot can be removed (e.g. page 11 line 52, p 11 line 30, etc.).* dots removed throughout.
- 3. The first time the mention to CURLA appears is in page 8, line 26, there its meaning must be provided, and conversely in page 16 line 4, only its acronym will suffice."*
..... corrected as recommended.

I think that this addresses all points raised in the third round of the Review process.

Mike Hands

December 30th 2020